# N2B27 media formulations influence gastruloid development

Tina Balayo[1,*], Sharna Lunn[2], Pau Pascual-Mas[1], Ulla-Maj Fiuza[1], Amruta Vasudevan[2], Joshua D. Frenster[1], Joel B. Josende García[1], Hannah Y. Galloon[2], Raquel Flores Peirats[1], Alfonso Martinez Arias[1,3], André Dias[1,*,‡] and David A. Turner[2,‡]

## ABSTRACT

Gastruloids are 3D aggregates of pluripotent stem cells grown in suspension culture that mimic many aspects of gastrulation and early axial elongation. The N2B27 basal medium in which mouse gastruloids are cultured can either be home-made (HM-N2B27) with materials of known origin, or commercially sourced (NDiff227), where the exact formulation is unknown. In this study, we examined whether these formulations resulted in significant differences in gastruloid development. Our results reveal that while both media enable standard gastruloid elongation, HM-N2B27 gastruloids initiate the elongation process earlier, have a higher number of cells and an increased anterior domain. Despite the maintenance of overall gene expression patterns, RNAseq analysis indicated differences in cell fate specification, with HM-N2B27 gastruloids exhibiting higher expression of spinal cord-related genes, while NDiff227 favours mesodermal differentiation. Furthermore, differential gene enrichment analysis suggests that changes in key signalling pathways underlie the differences between HM-N2B27 and NDiff227 gastruloids. These findings highlight the importance of basal media composition for gastruloid development, underscoring the need for careful media selection during *in vitro* engineering of stem cell-based embryo models.

KEY WORDS: Mouse embryonic stem cells, Gastruloids, N2B27, NDiff®227, Cell fate, Signalling and gastrulation

## INTRODUCTION

The development of an embryo is a highly organised process, with several factors, such as signalling factors, gene and cell regulatory networks (GRNs and CRNs, respectively), and the physical constraints of the system, regulating the allocation of distinct cell types and coordinating cell movements and tissue morphogenesis (Steventon et al., 2021; Tam and Behringer, 1997). Many of the mechanisms involved in early embryonic patterning have been derived from *in vivo* approaches; however, in the case of mammals,

especially human embryo development, there are a number of challenges, technical and ethical, that need to be overcome (Rossant and Tam, 2021; Rugg-Gunn et al., 2023).

*In vitro* model systems, making use of pluripotent stem cells (PSCs), provide an alternative and tractable experimental approach to study these developmental processes (Fu et al., 2021; Steventon et al., 2021), and have become useful tools for understanding the process of cell fate decision-making in a physiological context. When grown in 3D and exposed to the appropriate signals, they can generate organoids that can be used to study the formation of specific tissue types and 'organ-like' structures (e.g. optic cup, gut and cerebral organoids) (Eiraku and Sasai, 2012; Sasai, 2013; Sasai et al., 2012). Gastruloids, which are embryonic organoids (i.e. stem cell-based embryo models), are made from small numbers of aggregated PSCs (Turner et al., 2016; Turner and Martinez Arias, 2024). When they are exposed to a short pulse of Wnt/β-catenin signalling after 48 h of culture in differentiation media, they mimic many aspects of the early gastrulating embryo, such as symmetry-breaking with a polarisation of posterior-related gene expression, axial elongation, and the formation of all three embryonic axes with collinear expression of Hox genes along the anteroposterior axis (Beccari et al., 2018; Dias et al., 2025b preprint; Turner et al., 2017, 2014a; van den Brink et al., 2014). A key feature of the gastruloid model system lies in its high reproducibility, which is crucial for the understanding of developmental processes (Merle et al., 2024; Triandafillou et al., 2025 preprint; Turner et al., 2017). It is therefore important to understand and properly characterise the factors that guide their differentiation, ensure reproducibility between experimental replicates, and acknowledge the variables that lead to suboptimal or alternative and/or variable phenotypes (Rosen et al., 2022 preprint; Turner and Martinez Arias, 2024; Villaronga-Luque et al., 2025). Recent work has shown that factors such as 2D culture conditions of the starting cell population (Blotenburg et al., 2025), initial number of cells (Anlas et al., 2024; Bennabi et al., 2025; Fiuza et al., 2025; Hamazaki et al., 2024; van den Brink et al., 2014), cell heterogeneity (Ayyappan et al., 2025 preprint; Regalado et al., 2025 preprint), signalling (Dias et al., 2025a preprint; Wehmeyer et al., 2025 preprint), metabolism (Dingare et al., 2024; Stapornwongkul et al., 2025; Villaronga-Luque et al., 2025) and molecular processes like post-transcriptional regulation (Taborsky et al., 2025 preprint) play a crucial role in gastruloid variability and can have a significant impact on fate specification. A critical element of all these variables is the basal medium in which cells and gastruloids are grown: N2B27.

The N2B27 media was initially developed as a chemically defined neural differentiation media (Mulas et al., 2019; Ying and Smith, 2003; Ying et al., 2003); however, due to the minimal nature of its composition, it served as a useful base in which to add key signalling components to direct PSCs (either in monolayer or as gastruloids) to distinct lineages in a controlled manner (for examples, see Gouti et al., 2014; Hennessy et al., 2023 preprint;

[1]Department of Medicine and Life Sciences, Universitat Pompeu Fabra, Barcelona 08003, Spain. [2]Institute of Life-Course and Medical Sciences, Faculty of Health and Life Sciences, University of Liverpool, Liverpool L7 8TX, UK. [3]ICREA, Passeig de Lluís Companys, 23, L'Eixample, 08010 Barcelona, Spain.
*These authors contributed equally to this work

‡Authors for correspondence (andre.dias@upf.edu; david.turner@liverpool.ac.uk)

T.B., 0000-0002-2185-6989; S.L., 0000-0002-6497-6649; P.P.-M., 0000-0002-6873-6538; U.-M.F., 0000-0002-8691-0325; A.V., 0000-0002-5777-9508; J.D.F., 0000-0003-2401-1575; J.B.J.G., 0009-0007-5442-7375; H.Y.G., 0009-0007-7930-2130; R.F.P., 0000-0002-5844-470X; A.M.A., 0000-0002-1781-564X; A.D., 0000-0003-3337-6373; D.A.T., 0000-0002-3447-7662

Turner et al., 2014a,b,c; van den Brink et al., 2014). N2B27 is primarily made from a 1:1 mixture of DMEM:F12 and Neurobasal media with added N2 and B27 (containing vitamin A) supplements (Mulas et al., 2019). A commonly used commercial version of this medium is known as NDiff®227 (NDiff227) and, presumably, contains similar components, although it is not possible to determine the exact quantities or makeup of this proprietary medium. Anecdotal evidence from several groups has suggested that gastruloid development may differ depending on whether one prepares the media in the lab or uses the commercial version. However, to our knowledge there has not been a systematic analysis of the differences between the two media formulations in the context of gastruloid development.

Here, we have undertaken a short study to quantitatively assess the effect of using a home-made N2B27 media formulation (HM-N2B27) or NDiff227 on gastruloid development, morphology and gene expression. We found that although the shape of the gastruloids was broadly consistent, there were significant changes in morphometric variables, such as the length and elongation index of the gastruloids. We also observed significant differences in cell number over time, with HM-N2B27 gastruloids consistently showing higher cell counts compared with their NDiff227 counterparts. In addition to a delay in the switch from Cdh1 (E-cadherin) to Cdh2 (N-cadherin) around 96 h (Basilicata et al., 2016; Mayran et al., 2023 preprint; Suppinger et al., 2023), we also detected differences in cell fates. At 120 h, HM-N2B27 gastruloids showed an increased expression of spinal cord-related genes, whereas NDiff227 gastruloids displayed higher levels of paraxial mesoderm-associated genes. A broader analysis of gene ontology (GO) and biological processes suggests that changes in key signalling pathways could underlie these observed differences in the gastruloids cultured with the two media. Overall, our work indicates that, while these distinct N2B27 media formulations do not affect the reproducibility of the gastruloid model system, they impact gastruloids at both the cellular and molecular levels, and suggest that the choice of media should be a key consideration in the experimental design.

## RESULTS AND DISCUSSION
### Distinct N2B27 media formulations impact gastruloid development, morphology and cell number
To understand if different N2B27 differentiation media formulations affect the development of mouse gastruloids, we set up several independent comparative experiments using various batches of commercial NDiff227 and home-made N2B27 (HM-N2B27; see Materials and Methods), and observed the morphology of wild-type E14Tg2A gastruloids at 120 h following the standard 48-72 h CHIR pulse – these experiments were carried out using cells pre-cultured in ESLIF (ESL) media containing 10% serum and following the standard CHIR gastruloid protocol, with a seeding density of 300 cells (see Materials and Methods). Although the classical elongation (Beccari et al., 2018; Turner et al., 2017; van den Brink et al., 2014) was obtained using both types of differentiation media, we observed it was higher when HM-N2B27 was used (~90%, against only around 80% with NDiff227; Fig. 1A,B). In addition, we noticed several clear differences regarding gastruloid morphology and the frequency of either single or multi axis gastruloids (Fig. 1). In the first instance, we observed that HM-N2B27 gastruloids had ~60% single axis elongations (Fig. 1A,B), whereas the use of NDiff227 resulted in a much higher frequency of single axis elongations (>90%; >100 individual gastruloids; from five independent biological replicate experiments; Fig. 1A,B). We also observed that HM-N2B27 gastruloids were greater in overall size (area) and displayed a more pronounced elongation in comparison to gastruloids developed using

NDiff227 (Fig. 1A,C). This trend was also maintained when gastruloids were made with E14Tg2A cells from different pre-culture conditions (e.g. 2iL or 15% serum ESL; Fig. S1A,B,E), and with gastruloids made with cells from a different genetic background (Fig. S1C). Notably, these differences between HM-N2B27 and NDiff227 gastruloids are likely not related to the aggregation process per se, because they were also present despite the use of ultra-low adhesion plates (see Materials and Methods), which facilitate cell aggregation (Fig. S1D,E).

The three gastruloid morphological measurements performed in this study – length, area and elongation index (EI) – are potentially connected. For example, if gastruloid length were to increase, but the proportionality of the gastruloid remained consistent, area would also increase. In addition, the EI is calculated as a ratio between the diameter of the largest inscribed circle within the gastruloid (which would relate to area) and the gastruloid total length (Fig. 1D) (Guiet et al., 2021). To consider these variable interactions, which could obfuscate results and lead to misinterpreted conclusions (Yu et al., 2022) (and S.L., A.V., H.Y.G., A.D. and D.A.T., unpublished), mixed effects models (MEMs) (Yu et al., 2022) were implemented to compare the morphologies of single axis gastruloids (10% ESL E14Tg2A) from both media (Fig. 1Ci,Cii and Fig. S2). Prior to the application of MEMs, observations of the raw data suggested that HM-N2B27 gastruloids had greater length and area than their NDiff227 counterparts, with no clear distinction in elongation index. However, after application of MEMs, HM-N2B27 gastruloids had a greater overall length ($P=1.24\times10^{-4}$) while NDiff227 gastruloids had a greater elongation index ($P=7.95\times10^{-4}$) and the observed difference in area was not significant (Fig. 1Cii, Fig. S2 and Table S1).

Reviewing the wide-field images of gastruloids using MEM analysis supports a more nuanced conclusion on gastruloid morphology than could be drawn from analysis of the raw data. HM-N2B27 have a larger anterior pole relative to their posterior compartment, which contributes to an increased overall length but reduces its elongation index. On the other hand, NDiff227 gastruloids display a more uniform shape from pole to pole, which results in a larger elongation index (Fig. 1A,C, Fig. S2 and Table S1). In addition, although the raw area data of single axis gastruloids are significantly different between the two formulations, once the influence of other variables was mitigated, this difference was attributed to the length, elongation index and proceeding interactions (Fig. 1Cii, Table S1). Measurements of the overall area of multi axis gastruloids also indicate that use of HM-N2B27 media can result in gastruloids with a greater area than NDiff227 ($P<0.0001$; Fig. 1Cii). However, in this case the length and elongation index of multi axis gastruloids could not be determined, as the longest axis to measure was often ambiguous. These results indicate that the N2B27 media composition impacts the development of CHIR-treated gastruloids, significantly affecting their morphology at 120 h, although their posterior elongation is maintained.

The observed size difference between 10% ESL E14Tg2A gastruloids grown in HM-N2B27 or NDiff227 led us to consider whether different media conditions directly influenced gastruloid cell number. To examine this, we quantified the number of cells in gastruloids from both culture media over time (Fig. 1E; Table S2). Owing to the limited number of cells at early time-points, multiple gastruloids were pooled together to improve count accuracy (see Materials and Methods for details). At each time-point, with the exception of 48 h, HM-N2B27 gastruloids showed consistent and significantly greater numbers of cells than NDiff227 gastruloids

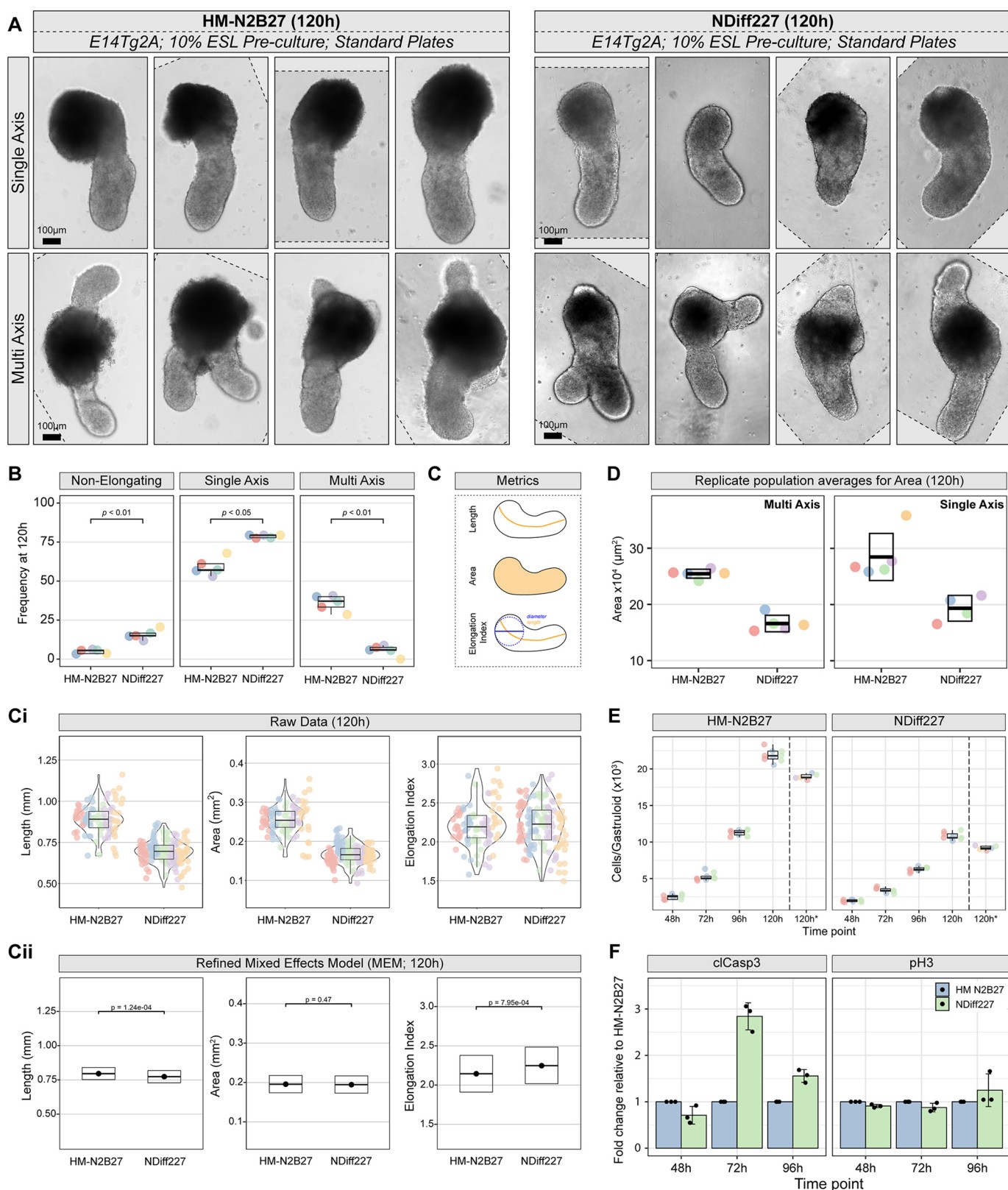

**Fig. 1.** See next page for legend.

(Fig. 1E; Table S2; $P<0.0001$). As averaging whole pools of gastruloids has the potential to introduce averaging errors, the cells from individual gastruloids ($n=5$) were also counted at the 120 h time-point (Fig. 1E; denoted 120 h*), and the results indicate

that HM-N2B27 gastruloids have a significantly higher, roughly double, number of cells than NDiff227 (Fig. 1E; Mann–Whitney test, $W=25$; $P=0.0079$). As several reports showed that increasing the number of seeding cells results in multi axis formation

**Fig. 1. HM-N2B27 and NDiff227 media significantly affect gastruloid morphology and cell number.** (A) Representative images selected from five independent experiments showing the general morphology of E14Tg2A CHIR-treated gastruloids at 120 h after aggregation, with either a single or multi axis grown in either HM-N2B27 (left) or NDiff227 (right). Scale bars: 100 μm. (B) Biological replicate population (pooled gastruloids, per experiment) averages for the frequency of non-elongating, single- or multi-axis gastruloids in HM-N2B27 and NDiff227 conditions. A Mann–Whitney *U*-test was used to compare gastruloids with non-, single- or multi-elongation in either HM-N2B27 or NDiff227. (C) Morphometrics analysis of gastruloids in A. Schematics indicate how each measurement was obtained. (Ci) Raw quantification of the length, area and elongation index of single axis gastruloids in HM-N2B27 or NDiff227. Individual gastruloids are represented as coloured circles, with distinct colours for each independent batch, and suggest an increase in the area and length of HM-N2B27 gastruloids in comparison to NDiff227 counterparts. (Cii) Predictive averages (±95% CI) generated from refined mixed effects models (MEMs) of data from at least 95 gastruloids per medium (NDiff227: 5 replicates, 150 gastruloids; HM-N2B27: 5 replicates, 153 gastruloids; see Materials and Methods, Fig. S2 and Table S1), highlighting significant differences in length and elongation index between HM-N2B27 and NDiff227 (Tukey-corrected post-hoc comparison). (D) Biological replicate population averages for the area of HM-N2B27 or NDiff227 gastruloids, indicating that HM-N2B27 gastruloids are bigger than their NDiff227 counterparts independently of having one or more axes. (E) Average number of cells in pooled E14Tg2A gastruloids from three independent experiments or from five independent gastruloids (120h*; right of hashed line) at the indicated time-points. Data indicate a significant increase in the number of cells in HM-N2B27 gastruloids in comparison to NDiff227 counterparts. See Table S2 for significance testing for pooled gastruloids. A Mann–Whitney *U*-test was used to compare the cell counts from individual gastruloids at 120 h (120 h*; *P*=0.0079). (F) Pools of gastruloids (see Materials and Methods) were dissociated at the indicated time-points and stained for either cleaved caspase 3 (clCasp3) or phospho-histone H3 (pH3). Data are presented as the fold change in the number of positive cells relative to the HM-N2B27 condition for each time-point for either clCasp3 (left) or pH3 (right). Although pH3 remains stable across all tested time-points, NDiff227 gastruloids display an almost threefold increase for clCasp3 at 72 h. The data presented are from three independent biological replicate experiments, with the readout made through flow cytometry (see Fig. S3 and Materials and Methods for the gating strategy and other details). For the box and whisker plots, the central line in the box is the median, the lower edge of the box is the first quartile, the upper edge of the box is the third quartile, and the box height is the IQR. The lower and upper whiskers indicate the smallest datapoint within 1.5×IQR below Q1 or above Q3, respectively.

(Bennabi et al., 2025; Fiuza et al., 2025; van den Brink et al., 2014), it is possible that the higher frequency of multi axes in HM-N2B27 gastruloids is linked to the increased number of cells in comparison to NDiff227 gastruloids. This proved to be the case, since reducing the number of cells used during aggregation of HM-N2B27 gastruloids resulted in a significantly lower amount of multi axis formation, with HM-N2B27 gastruloids formed with 150-200 cells displaying a similar percentage of multi axes than NDiff227 gastruloids developed with 300 cells (Fig. S1F).

Our result indicating that, when the number of seeding cells is kept consistent, HM-N2B27 gastruloids display a two-fold increase in cell number compared to NDiff227 suggests that the use of distinct N2B27 media may impact the survival or rate of cell proliferation during gastruloid development. To examine these possibilities, we took a flow cytometry approach to quantify the percentage of cleaved caspase 3 (clCasp3)-positive cells, a marker of potential apoptosis or high stress (Porter and Janicke, 1999), and phospho-histone H3 (pH3), a marker of cell proliferation (Kim et al., 2017), in fixed dissociated cells from pools of 10% ESL E14Tg2A gastruloids cultured in either HM-N2B27 or NDiff227 (Fig. 1F; Fig. S3). Specifically, at 72 h, cells from gastruloids cultured in NDiff227 showed an almost threefold increase in the percentage of clCasp3-positive cells relative to HM-N2B27 (Fig. 1F, left). Conversely, we detected no major differences in the levels of pH3 at any time-point between HM-N2B27 and NDiff227 (Fig. 1F, right). Our data therefore suggest that the lower cell numbers, and consequently the smaller sizes, found within gastruloids cultured in NDiff227 are probably due to increased cell death at earlier time-points, rather than to changes in cell proliferation promoted by the different media. Along the same line, a recent study suggested that cell competition, the process by which sub-optimal cells are eliminated, was reduced in gastruloids developed with HM-N2B27 media in comparison to NDiff227 (Frenster et al., 2025 preprint).

A further morphological difference was also found that related to the timing of gastruloid development: 10% ESL E14Tg2A NDiff227 gastruloids usually displayed a round shape at 96 h, some ovoid, and a small number of gastruloids with single or multiple protrusions were also found depending on the batch number (Fig. 2 and Fig. S4A). By contrast, HM-N2B27 gastruloids had significantly less roundness [Fig. 2A; unpaired *t*-test, *t*(4)=−4.9541, *P*=0.008] and a higher frequency displayed protrusions (Fig. 2B and Fig. S4A), suggesting that the process of elongation started earlier in the home-made condition. In agreement, we noticed that the cadherin switch from Cdh1 to Cdh2 (Basilicata et al., 2016; Mayran et al., 2023 preprint; Suppinger et al., 2023) started around 6 h earlier, with 90 h HM-N2B27 gastruloids showing an expression pattern more similar to 96 h NDiff227 gastruloids (Fig. 2C,D and Fig. S4B). Together, these results suggest that distinct N2B27 media formulations can impact gastruloid developmental time, size and morphology, and leave the question unanswered of whether germ layer specification and global gene expression patterns are also affected.

## N2B27 medium influences key signalling networks and cell fate specification in the gastruloid model system

To understand whether the differences in gastruloid development we observed in different media formulations were also reflected at the level of cell fate specification, we harvested 10% ESL E14Tg2A CHIR-treated gastruloids at 120 h using HM-N2B27 and NDiff227, and analysed their gene expression by *in situ* hybridisation chain reaction (HCR) (Fig. 3). On the one hand, we observed an overall similarity in terms of expression patterns. For instance, the caudal epiblast markers *Cdx2* and *Tbxt* (Dias and Aires, 2020; Wymeersch et al., 2021) were expressed in a localised domain at the posterior/caudal part of both types of gastruloid (Fig. 3), suggesting that the symmetry-breaking process resolving anterior/posterior cell identities (Beccari et al., 2018; Turner et al., 2017; van den Brink et al., 2014) had occurred independently of the N2B27 media. This is also supported by the expression of both *Sox2* and *Tcf15* (Fig. 3) in a similar manner to what is described in the literature (Beccari et al., 2018), highlighting the presence of neural and paraxial mesodermal derivatives, respectively. On the other hand, we noticed some differences in the amount of key fate-specific genes between the two N2B27 conditions. For example, *Kdr*, which is related to endothelial cell identities (Ragusa et al., 2025; Rossi et al., 2021, 2022), was upregulated in HM-N2B27 gastruloids at 120 h (Fig. 3). Similarly, we found that the levels of *Sox17* – a gene also expressed in endothelial progenitors (Rossi et al., 2022) – were increased in home-made versus NDiff227 gastruloids (Fig. 3). Another difference lies in the *Tcf15* expression domain, which is slightly more posterior in NDiff227 gastruloids (Fig. 3), suggesting the existence of more paraxial mesoderm-like tissues in these gastruloids. Given the above-mentioned differences in cell number between HM-N2B27 and

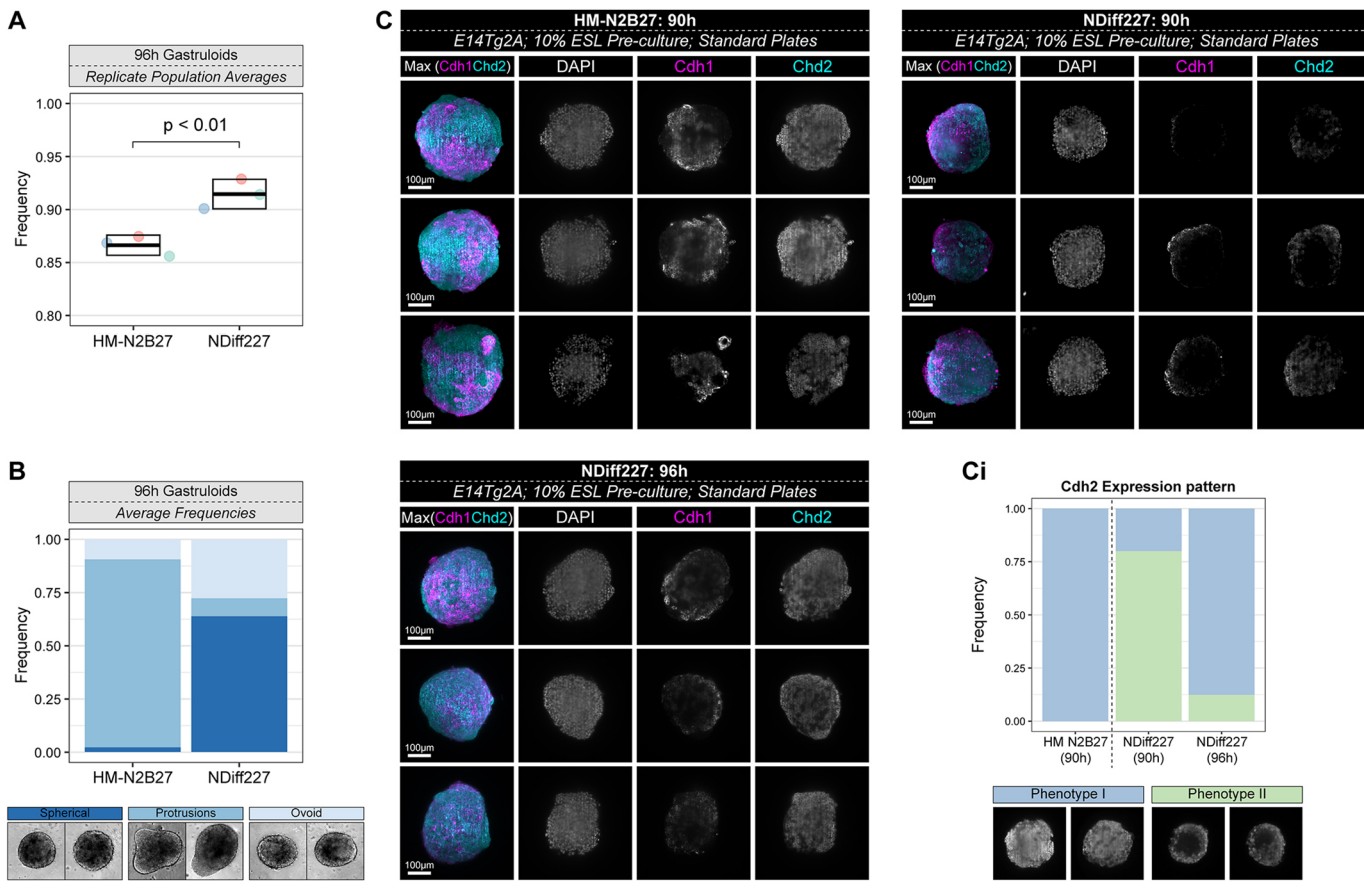

**Fig. 2. NDiff227 gastruloids display a developmental delay in comparison to HM-N2B27 gastruloids.** (A) Quantitative analysis of the roundness (perimeter$^2$/4π×area) of gastruloids cultured in either HM-N2B27 or NDiff227 showed that HM-N2B27 gastruloids are significantly less 'Round' than their NDiff227 counterparts at 96 h [unpaired two-tailed *t*-test; $t(4)=-4.9541$, $P=0.008$]. Each datapoint is the average roundness of an independent replicate experiment containing between ~40 and 90 individual gastruloids (see Fig. S4B). (B) The frequency of morphologies from the data in A, with examples of gastruloids that would fall in each category (bottom; the left 'Protrusions' image is also shown in Fig. S4). (C) Light-sheet representative images (maximum projection images) highlighting the expression patterns of Cdh1 (magenta) and Cdh2 (cyan) between HM-N2B27 gastruloids at 96 h and NDiff227 gastruloids at 90 h and 96 h. Gastruloids developed in HM-N2B27 have upregulated Cdh2 by 90 h, unlike NDiff227 gastruloids, which show enhanced Cdh2 only at 96 h. Scale bars: 100 µm. (Ci) Qualitative analysis of the average expression pattern of Cdh2 (N-cadherin). The expression patterns were grouped into two categories (phenotypes I and II) and represent the average of three independent replicate experiments, each containing at least seven individual gastruloids per condition. Examples of the two categories are shown below. The leftmost example for 'Phenotype I' is shown in the middle row of 90 h HM-N2B27 (Cdh2) in C. For the box and whisker plot, the central line in the box is the median, the lower edge of the box is the first quartile, the upper edge of the box is the third quartile, and the box height is the IQR. The lower and upper whiskers indicate the smallest datapoint within 1.5×IQR below Q1 or above Q3, respectively.

NDiff227 gastruloids (Fig. 1E), we questioned whether reducing the initial cell number in the homemade condition would resolve the differences observed at the level of gene expression. However, after HCR analysis of HM-N2B27 gastruloids developed from an initial pool of 200 cells (standard seeding number was 300 cells), we observed that the differences related to *Kdr*, *Sox17* and *Tcf15* were also largely present in these gastruloids (Fig. 3). Therefore, this suggests that cell number is likely not the main driver underlying the differences in gene expression between HM-N2B27 and NDiff227 gastruloids. In agreement, Bennabi et al. and Fiuza et al. also observed that transcriptional programs and cell fate composition in mouse gastruloids are stable within a size range (Bennabi et al., 2025; Fiuza et al., 2025).

Next, we took a bulk-RNAseq approach to examine quantitatively whether other transcripts showed differential expression when pools of gastruloids were developed, from the same number of cells (300, see Materials and Methods), using either NDiff227 or HM-N2B27. Statistical pairwise comparison showed that 196 genes were

differentially expressed [Log fold change (LFC)=1.5, *P*-value *P*<0.05] between HM-N2B27 and NDiff227 samples (Table S3; Fig. 4A). Principal component analysis (PCA) revealed that both samples form distinct separate clusters in the first dimension (accounting for 71% of the variance), with a considerable distance between the two (Fig. 4B). The top 100 genes based on loadings of PCA dimension 1 and 2 (Table S4; Fig. 4B) are in agreement with the HCR results and suggest that these differences might be associated with fate specification, particularly with different amounts of endothelial cells (e.g. *Kdr* and *Cdh5*), neuroectoderm (e.g. *Pax6* and *Pou3f2*) and presomitic/paraxial mesoderm (e.g. *Tbx6*, *Lfng* and *Tcf15*). Although both gastruloids made with either HM-N2B27 or NDiff227 developed the classical derivatives of the late/posterior PS (Dias et al., 2025b preprint; Wehmeyer et al., 2025 preprint), an in-depth analysis of specific fate markers indicated significant differences in the levels of genes associated with neuroectoderm and paraxial mesoderm. HM-N2B27 gastruloids exhibited higher levels of *Sox1*, *Sox2*, *Sox3* and *Pax6* than those made with NDiff227

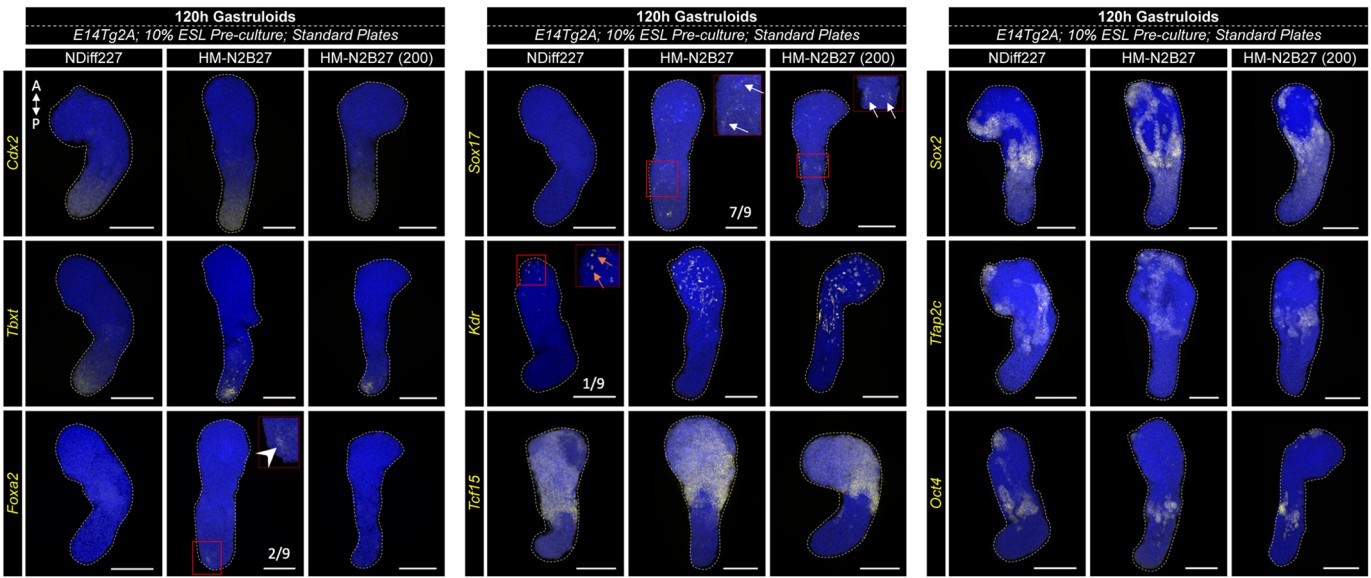

**Fig. 3. *In situ* HCR analysis comparing the expression of key lineage markers in NDiff227 and HM-N2B27 gastruloids at 120 h.** Gastruloids cultured in NDiff227 or HM-N2B27 (data are representative of three independent biological replicate experiments, with a minimum of three gastruloids per replicate) were probed for genes associated with the caudal epiblast (*Cdx2* and *Tbxt*), notochord (*Tbxt* and *Foxa2*), endoderm (*Foxa2* and *Sox17*), endothelial (*Sox17* and *Kdr*), paraxial mesoderm (*Tcf15*), neural tube (*Sox2*) and those associated with Gld-PGCLCs/ectopic pluripotency (*Sox2*, *Tfap2c* and *Oct4*). HM-N2B27 gastruloids were developed with both 300 (standard, similar to NDiff227 gastruloids) or 200 cells [HM-N2B27 (200)]. The assessed gene expression patterns are comparable across all conditions, although there are some minor differences. For example, the *Tbxt* domain seems smaller in the HM-N2B27 conditions. Conversely, *Kdr* expression is increased when home-made media was used, regardless of the starting number of cells (*Kdr* expression was detected only in one out of nine NDiff227 gastruloids, orange arrows). In agreement, *Sox17* was frequently observed in the two types of HM-N2B27 gastruloids (white arrows). Sporadic expression of *Foxa2* was detected in standard HM-N2B27 gastruloids (two out of nine, white arrowhead), but not in the other conditions. In addition, the *Tcf15* expression domain appears more posteriorized in NDiff227 gastruloids. No major differences were observed regarding *Sox2*, *Tfap2c* and *Oct4*. Insets show higher magnifications of the regions of interest outlined in red. A, anterior; P, posterior. Scale bars: 200 µm.

(Fig. 4C). These differences in neuroectoderm are associated with spinal cord-like cells and not with brain-like structures (Girgin et al., 2023; Pfister et al., 2007), as there was no significant expression of *Hesx1*, *Pou3f1* and *En1* (Fig. 4C). On the other hand, gastruloids made with NDiff227 expressed higher levels of *Tbx6*, *Dll1*, *Pcdh19*, *Lfng*, *Tcf15* and *Raldh2* (*Aldh1a2*) than those made with HM-N2B27 (Fig. 4C). These differences are associated with an increase in early/nascent, presomitic and paraxial mesoderm-like tissues (Dias and Aires, 2020; van den Brink et al., 2020). Interestingly, we found that several Notch-related genes, such as *Notch1*, *Dll1* (Geffers et al., 2007), *Dll3* (Chapman et al., 2011), *Hes7* (Kageyama et al., 2007), *Mesp2* (Barrantes et al., 1999; Takahashi et al., 2000), *Runx1* (Li et al., 2018; Zhou et al., 2022) and *Nrarp* (Krebs et al., 2012; Lamar et al., 2001) were differentially regulated between HM-N2B27 and NDiff227 gastruloids at 120 h (Fig. S5). Given the central role that Notch plays in the differentiation of mammalian axial progenitors towards the neural and mesodermal lineages (Cooper et al., 2024; French et al., 2024 preprint; Gray and Dale, 2010), it is possible that Notch signalling activity might vary depending on the N2B27 media used and that it drives the observed differences in neural and mesodermal-related gene expression between the two types of gastruloids. Testing this hypothesis would require further mechanistic studies involving systematic gain- and loss-of-function experiments, along with the monitoring of both Notch signalling activity and the expression of neural and mesodermal marker genes. Finally, we also found no major differences in lateral/intermediate mesoderm [e.g. *Lhx1*, *Osr1* and *Pax2* (Beccari et al., 2018; Dias et al., 2020; Dressler, 2009)], endoderm [e.g. *Foxa2 and Apela* (Beccari et al., 2018; Nowotschin et al., 2019; Pfister et al., 2007)] or notochord-related gene expression (Beccari et al., 2018; Wymeersch et al., 2019) (e.g. *Noto and Shh*; Fig. 4C).

Euclidean distance analysis between the two types of gastruloid and mouse embryo (Pijuan-Sala et al., 2019) indicated that the development of the gastruloids occurred as expected, reaching early stages of axial elongation (Beccari et al., 2018), with gastruloids made with NDiff227 being more related to E8.25 mouse embryos, whereas those obtained with HM-N2B27 seemed more similar to E8.5 embryos (Fig. 4E). Concomitantly, the expression of Hox genes was found to not differ substantially between the two types of gastruloids (Fig. 4D). Small differences were observed in Hox genes that, at these stages, are known to extend up to more anterior regions of the body axis (*Hoxc4*, Hoxc5, *Hoxc6* and *Hoxb9*) (Deschamps and Duboule, 2017; Goh et al., 1997; Minguillon et al., 2012; Nishimoto et al., 2014). These Hox genes were slightly upregulated in HM-N2B27 gastruloids (Fig. 4D), suggesting they might contain more anterior-like structures. In agreement, we found that some genes associated with endothelial cells were upregulated in gastruloids made with HM-N2B27 (e.g. *Kdr* and *Cdh5*; Fig. 4C). On the other side, NDiff227 gastruloids displayed higher levels of *Hoxd9*, which is more restricted to the caudal part of embryo, and of genes like *Tbxt*, *Nkx1-2*, *Cdx2* and *Cdx1* (Fig. 4C,D), which are associated with the neuromesodermal competent population (Binagui-Casas et al., 2021; Wymeersch et al., 2021). The increase of *Hoxa5* in these gastruloids (Fig. 4D) is likely due to the increase in mesodermal-like structures (Minguillon et al., 2012). A further key difference between NDiff227 and HM-N2B27 gastruloids is that the former contains significantly higher levels of E-cadherin (*Cdh1*), *Oct4* (*Pou5f1*), *Nanog*, *Utf1*, *Rex1* (*Zfp42*) and *Tfap2c* (Fig. 4C). This expression signature is similar to that reported for gastruloid-derived primordial germ-like cells (Gld-PGCLCs) (Cooke et al., 2023; van den Brink et al., 2020) or, at this stage in mouse gastruloids, ectopic pluripotency (Suppinger et al., 2023). Given the

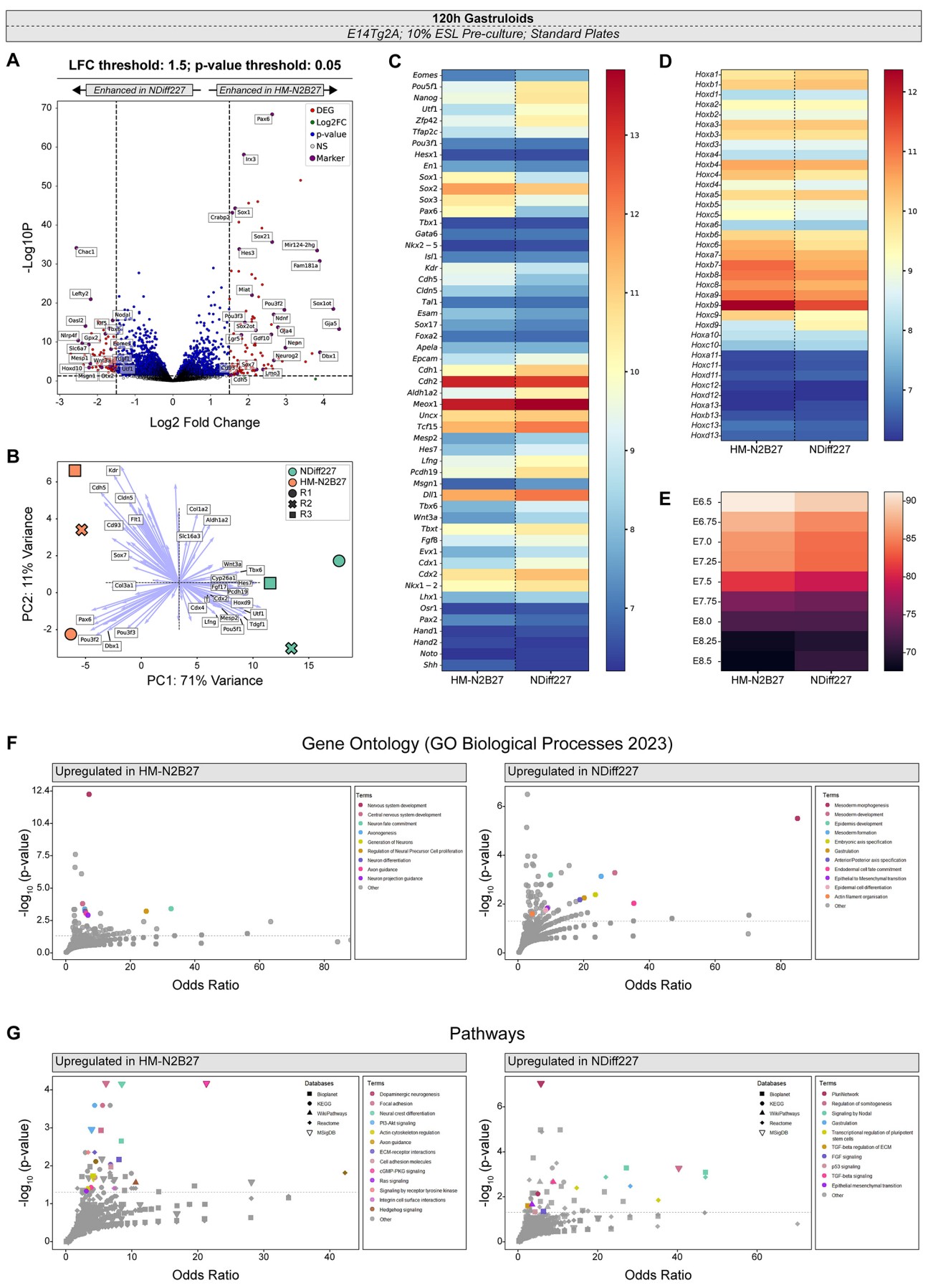

**Fig. 4.** See next page for legend.

**Fig. 4. Transcriptomic analysis between HM-N2B27 and NDiff227 gastruloids.** Bulk RNAseq analysis of pooled 10% ESL pre-cultured E14Tg2A gastruloids harvested at 120 h, cultured in either HM-N2B27 or NDiff227 (three biological replicates per condition). (A) Volcano plot of differentially expressed genes (DEGs) with a log-fold change (LFC) threshold of 1.5 (indicated in red) and significance threshold ≤0.05 (blue and red). Specific DEGs of interest are highlighted, and the full list can be found in Table S3. (B) Principal component analysis (PCA), with some of the top 100 genes based on loadings of PCA dimensions 1 and 2 (purple arrows; full gene list can be found in Table S4), highlighting the differences between NDiff227 (in green) and HM-N2B27 gastruloids (in orange; three biological replicates per condition). (C) Heatmap of selected fate marker genes (mean of the replicates variance-stabilised read counts), highlighting the differences and similarities between HM-N2B27 and NDiff227 gastruloids at 120 h. There is a difference in the expression of spinal cord (*Sox1*, *Sox2*, *Sox3* and *Pax6*) and paraxial mesoderm-related genes (*Aldh1a2*, *Meox1*, *Uncx* and *Tcf15*) between the two media conditions. In addition, genes related to axial progenitors (e.g. *Tbxt*, *Cdx2* and *Nkx1-2*) are upregulated in NDiff227 gastruloids. (D) Comparative heatmap (mean of the replicates variance-stabilised read counts) showing minor differences related to Hox gene expression. (E) Euclidean distance analysis between gastruloids cultured in the two media conditions and mouse embryos, from embryonic day (E) 6.5 to E8.5, highlights a similar temporal development between the two. (F,G) Over-representation analysis (ORA) conducted on the DEGs (set using *P*<0.05, and an absolute log2 fold change>1 – see Materials and Methods) at 120 h, highlighting differences in biological processes (F) and signalling pathways (G) between the two media conditions. All datapoints represent individual annotation terms from the GO Biological Processes 2023 database (G) and the pathways databases (F), with each shape pertaining to a unique database (refer to 'Databases' in the keys in G). The significantly enriched and biologically meaningful datapoints are coloured according to the annotation terms. The *x*-axis is the Odds Ratio, the y-axis is -log$_{10}$ (*P*-value adjusted for multiple testing) and the dotted line corresponds to an adjusted *P*-value of 0.05.

overall downregulation of these genes in HM-N2B27 gastruloids, it is likely that the differences imposed by the N2B27 media are not related to the increase of one cell identity over the other (PGCLCs versus ectopic pluripotency), but rather due to a slight difference in the proportion of these cells in the overall gastruloid. This becomes apparent from the well-defined, and similar, expression domains of *Oct4* and *Tfap2c* in our HCR analysis (Fig. 3).

A broader analysis (LFC=1, *P*-value *P*<0.05) that was focused on signalling pathways, gene ontology and biological processes also revealed significant differences between HM-N2B27 and NDiff227 gastruloids at 120 h (Fig. 4F,G). In addition to an enrichment of pluripotency and mesoderm-related processes, our results suggest that the expression of genes related to the FGF, p53 and TGFβ/Nodal signalling pathways is enhanced in gastruloids made with NDiff227 (Fig. 4F; Table S5). For example, the enrichment of the last is a consequence of the upregulation of both Nodal and its downstream targets, *Lefty1* and *Lefty2* (Bisgrove et al., 1999; Meno et al., 1999), in NDiff227 gastruloids (Fig. S4; Table S5). In contrast, PI3K-Akt, cGMP-PKG, Ras, and Hedgehog signalling were enriched in HM-N2B27 gastruloids (Fig. 4F, top; Table S5). The PI3K/Akt signal cascade is a known regulator of several key cellular functions such as growth, migration, differentiation and cell survival (Riley et al., 2005). Recent work in gastruloids has shown that this signalling pathway is required for normal axis elongation (its inhibition reduces gastruloid length) and can induce proliferative activity in the anterior compartment of gastruloids, leading to its expansion (Underhill and Toettcher, 2023). These results are consistent with our observations as HM-N2B27 gastruloids have an enrichment of PI3K/Akt-related genes (e.g. *Igf1*, *Ccnd2*, *Thbs1* and *Itgb8*) (Scrima et al., 2012; Shu et al., 2019; Stitt et al., 2004; Tuoheti et al., 2024) and display an increased anterior

compartment and length in comparison to NDiff227 gastruloids (Fig. 1 and Fig. S5, Table S5). Finally, our GO analysis also suggested an enrichment of cellular processes related to cell adhesion/motility, extracellular matrix and cytoskeleton regulation due to the upregulation of *Mylk* (*Mlck*) (Blaser et al., 2006; Cai et al., 1998; Samson et al., 2019; Weiser et al., 2009), *Nrcam* (Grumet et al., 1991), *Lama4* (Malinda and Kleinman, 1996), *Col15a1* (Eklund et al., 2001; Rasi et al., 2010), *Col3a1* (Kuivaniemi and Tromp, 2019) and *Thbs1* (Murphy-Ullrich, 2019) in HM-N2B27 gastruloids (Fig. S4, Table S5). In various studies (Cermola et al., 2022; de Jong et al., 2024; Fiuza et al., 2025; Mayran et al., 2023 preprint; Pineau et al., 2025; Veenvliet et al., 2020) these processes have been shown to be crucial for gastruloid elongation. For instance, forced reduction in actin polymerisation through the use of cytochalasin D between 90-120 h resulted in a more prominent elongation in mouse gastruloids treated with CHIR (Fiuza et al., 2025). Therefore, our work provides an extensive list of pathways and target genes that can be used in subsequent studies aimed at identifying and modulating factors that play a role during axial elongation, both *in vivo* and *in vitro*. As the gastruloid model system is not confined to mouse ESCs, and human gastruloids have recently been developed (Moris et al., 2020), it will be interesting in the future to address whether the human system is also sensitive to distinct media formulations and to understand the biological implications of such results.

## Conclusions
Our findings demonstrate that variations in N2B27 media formulations can significantly influence key aspects of mouse gastruloid development. These include gastruloid morphometrics, developmental timing, gene expression, cell fate specification, signalling networks and likely also the rate of cell death. Therefore, we propose that the N2B27 media (and other similar differentiation media) should be considered an important experimental variable in gastruloid studies and, more broadly, in the *in vitro* engineering of stem cell-based embryo models.

## Limitations
Owing to the specific nature and scope of this study, there are limitations that a more in-depth, mechanistic analysis could address in the future. One limitation is the narrow assessment of inter-gastruloid transcriptomic variability between the two media conditions, especially across different batches. Addressing this issue would require a more exhaustive study, involving scRNAseq and a higher number of inter- and intra-batch gastruloids assessed via HCR and immunofluorescence. Another limitation has to do with understanding whether, and if so how, the cellular and/or morphological changes in gastruloids cultured in the two media are linked to the observed genetic differences. This would require further systematic work beyond the question we set out to evaluate in this study – determining whether commercial and home-made N2B27 media formulations function similarly in mouse gastruloid development. A third limitation regards the lack of assessment on whether the activity of the potential differentially regulated signalling pathways was indeed affected by the two media compositions and whether the differentially expressed genes associated with processes such as cell adhesion and/or motility, extracellular matrix organization, and cytoskeletal regulation are functionally relevant for mouse gastruloid development. Addressing these questions would require specific reporter cell lines, which are unavailable in our laboratories, as well as mutagenesis or pharmacological treatments during gastruloid development. While we aim to conduct some of these experiments in

the future, they fall beyond the scope of this work. Finally, another limitation of our study relates to our inability to properly compare the ingredients of the two N2B27 media compositions due to the proprietary nature of the commercial NDiff227 recipe.

## MATERIALS AND METHODS

### N2B27 media formulations

Commercial NDiff®227 (NDiff227) was obtained from Takara Bio (Takara, Y40002) and the following lots were used in this study: 'AM30020S', 'AM10016S', 'AM90020S', 'AN30020S', 'AO30013S' and 'AOZ0020S'. NDiff227 media displaying a non-uniform colour is not optimal for gastruloid culture and, therefore, was not used in this study. Home-made N2B27 (HM-N2B27) was prepared in lots of 50 ml by mixing equal volumes of freshly made N2 and B27 media, together with β-mercaptoethanol (Invitrogen, 31350010) at a concentration of 1:1000. 25 ml of N2 media were prepared using 24.5 ml of DMEM/F12 (1:1) (Gibco, 21331-020), 250 µl of N-2 Supplement (100×; Gibco, 17502-048; Lot: '2584689', '2584685', '2831191' and '2868559') and 250 µl of L-Glutamine (Invitrogen, 25030-024). 25 ml of B27 media were prepared using 24.25 ml of Neurobasal (Gibco, 21103-049), 500 µl of B27 Supplement (50×; Gibco, 17504-044; Lot: '2596518', '2596510', '2814925' and '2954053') and 250 µl of L-Glutamine. HM-N2B27 used to develop gastruloids from mouse ESCs pre-cultured with 2iLIF media (2iL, see below) was prepared in the Liverpool laboratory (standard was prepared in Barcelona), following a similar recipe but some ingredients were acquired from different suppliers, batches or catalogue numbers: DMEM/F12 (1:1) (Gibco; 11320074), N2 supplement (Gibco; 175020-48; Lot: '2868549' and '2868551'), B27 supplement (50×) (Gibco, 17504-044; Lot: '2814935', '2814937' and '2831197') and Glutamax (ThermoFisher, 35050038) was used instead of L-Glutamine. The N2 Supplement, B27 supplement and L-Glutamine were kept at −70°C and taken to 4°C only the day before the preparation of HM-N2B27. Similarly to NDiff227, HM-N2B27 was also protected from light and stored at 4°C for no more than 2 weeks. The storage conditions are crucial, especially if L-Glutamine is used instead of Glutamax, as improper storage will lead to its degradation (Jagusic et al., 2016; Mulas et al., 2019). Both types of N2B27 media were batch tested before the experiments described in this study to assess their ability to generate gastruloids and the degree of morphological inter- and intra-batch variation. Table S6 summarises the parameters that are critical for N2B27 and NDiff227 storage, preparation and quality control.

### Cell lines and gastruloid development

Wild-type E14Tg2A (Hooper et al., 1987) and R1 H2B::mCherry;GFP-GFP (Nowotschin et al., 2009) mouse embryonic stem cells (ESCs) were maintained in the Barcelona lab in 10% (standard) or 15% (high-serum condition) ESLIF (ESL) media as indicated by Dias et al. (2025b) and Frenster et al. (2025), respectively. Wild-type E14Tg2A and Bra::GFP (Fehling et al., 2003) were cultured in the Liverpool lab for at least 2 weeks in 2iL [N2B27 base medium supplemented with 3 µM CHIR9901 (CHIR; Tocris), 1 µM PD03 (Tocris; 4192) and 11 ng/ml of LIF (Qkine; Qk018)] before the gastruloid protocol, which in all tested conditions used 3 µM CHIR (Sigma, SML1046, Barcelona lab; 4423, Tocris, Liverpool lab) between 48 and 72 h, as previously described (Beccari et al., 2018; Turner et al., 2017; van den Brink et al., 2014). The standard number of cells used to make gastruloids was 300, but some experiments were performed with 150 and 200 cells. Non-adherent U-bottomed 96-well plates (Greiner, 650185) were used for most experiments, and ultra-low attachment plates (Greiner, 650970) were applied only in certain circumstances to favour the aggregation process. The outer wells of 96-well plates were filled with 150 µl of PBS to minimise evaporation of the gastruloid culture medium. Cells were counted using a Countess Automated Cell Counter 3 (Invitrogen; Barcelona lab) or a TC20 automated cell counter (BioRad; Liverpool lab). Gastruloids were developed in a 37°C, 5% $CO_2$ incubator using HM-N2B27 or NDiff227 media. Every addition of the media between and including 48 h after aggregation and 96 h after aggregation was administered with vigorous agitation to ensure limited attachment of gastruloids to the well surface. Images of gastruloids were taken in Barcelona using an Olympus brightfield CKX53 microscope (4× and 10× lenses), equipped with an EP50 camera, or a Zeiss Cell Observer fluorescence microscope (5× or 10× lenses

depending on the size of the gastruloid), and in Liverpool using a Nikon Ti-E inverted widefield microscope (set to 37°C, 5% $CO_2$) using a 10×0.3 NA objective with data capture using the Nikon NIS-Elements AR software. Gastruloids that did not elongate were removed from the analysis, which equated to ∼10% with HM-N2B27 and ∼20% with NDiff227, depending on the batch of media used. See Table S2, which has additional troubleshooting information on stem cell culture and gastruloid development.

### Statistical analyses and data presentation

Statistical analysis for Figs 1B,D,E, 2A,C and Fig. S1F was performed using RStudio version 2024.12.1+563. Prior to statistical analysis, data were tested for normality (Shapiro-Wilk) and equal variance (either the F-test, Bartlett's test or through R performance package). After MEMs determined significant variation among conditions, a pairwise Tukey-corrected post-hoc comparison was used to determine significant differences between mediums of each individual morphological metric. $P<0.05$ was considered statistically significant. All graphs were generated using either RStudio or Prism software (Graphpad).

### Mixed effects models

Mixed-effects models used for morphological analysis of area, length and elongation index (Fig. 1Cii) considered media, area, elongation index and length as fixed variables, while biological replicates (i.e. independent gastruloid differentiations) were included as a random effect in a random intercept model, and random slope models were used where appropriate (Yu et al., 2022). Each model was optimised to fit statistical assumptions including homoscedasticity and normal distribution of residuals; performance of each model can be found in Fig. S2A-C. Graphical representation of the mixed effects models, in Fig. 1Cii, demonstrates mean predictive data and 95% confidence intervals generated from the *predictInterval* function using 1000 simulations of each model. After accommodating for interactions between fixed variables, Tukey's post-hoc pairwise comparisons were used for each morphological characteristic to determine statistical significance between HM-N2B27 and NDiff227.

### Gastruloid cell counting

Standard E14Tg2A gastruloids (10% ESL preculture) grown in HM-N2B27 or NDiff227 were pooled, washed and used for cell counting (Countess 3 Cell Counters, Invitrogen) at different stages according to the following quantities: 48 h, 92 gastruloids; 72 h, 46 gastruloids; 96 h, 24 gastruloids; 120 h, 12 gastruloids. Dissociation was carried out with Accutase (Capricorn Scientific, ACC-1B), in accordance with Dias et al. (2025b). Three independent experiments were performed and five gastruloids for each condition at 120 h were also dissociated and their cells counted in a separate manner.

### Immunofluorescence stains for flow cytometry

Flow cytometry quantification of cleaved-caspase 3 (clCasp3) and phospho-histone 3 (pH3) was conducted in three independent experiments, as described by Frenster et al. (2025). In brief, for each experiment, two 96-well plates each of gastruloids at 48 h and 72 h, and one 96-well plate of gastruloids at 96 h were collected and pooled, respectively. Gastruloids were dissociated with Accutase (LabClinics, ACC-1B) and fixed with 4% formaldehyde. After permeabilisation with 0.1% Triton X-100 and blocking in 10% bovine serum albumin solution, cells were stained with the primary antibodies anti-phospho-histone 3 (dilution 1:1600, Cell Signaling, 3377) and anti-cleaved-caspase 3 (dilution 1:5000, Cell Signaling, 9664). A goat-anti-rabbit Alexa-488-conjugated secondary antibody was used to separately visualise the primary antibodies (dilution 1:1000, Invitrogen, A-11034). Samples were analysed using a BD Bioscience LSRFortessa system. Cells were pre-gated against debris using FSC-A versus SSC-A and duplets were excluded using FSC-A versus FSC-H and FSC-A versus FSC-W. Unstained and secondary antibody-only controls from the same cell samples were used to set gates for true positive events (see Fig. S2 for the gating strategy).

### Immunofluorescence staining for light sheet imaging

Immunohistochemistry stains in individual whole-mount 10% ESL pre-cultured E14Tg2A 90 and 96 h gastruloids (eight gastruloids per condition,

three independent replicate experiments) were carried out in accordance with Fiuza et al. (2025). The primary antibodies used were goat anti-E-cadherin (AF648, R&D Systems; 1:500) and rabbit anti-N-cadherin (ab18203, Abcam; 1:200). The secondary antibodies used were donkey anti-rabbit Alexa Fluor 647 (A31573, ThermoFisher; 1:500) and donkey anti-goat Alexa Fluor 488 (A11055, ThermoFisher; 1:500). Immunostained gastruloids were imaged with a Viventis LS1 light-sheet microscope using opposing dual illumination (10× objective/NA 0.3) from the sides and single detection (25× objective/NA 1.1) from below and an exposure time of 50 ms. The signal was captured using a Hamamatsu ORCA-Fusion Digital CMOS camera (C14440). For illumination, 405 nm, 488 nm and 638 nm laser Gaussian beams of ∼3.3 µm were used with the appropriate band-pass filters for detection [respectively, CFP (BP 483/40), GFP/mCherry (DBP) and GFP/mCherry/638 (TBP)]. Gastruloid z-stacks were generated using a step of 5 µm between focal planes. Data analysis was performed in the ImageJ package Fiji (Rueden et al., 2017; Schindelin et al., 2012) and Imaris (Bitplane).

### *In situ* hybridization chain reaction
HCR was performed in accordance with Dias et al. (2025b) using E14Tg2A 120 h HM-N2B27 and NDiff227 gastruloids developed using 200 or 300 cells, pre-cultured in 10% ESL. HCR probes and amplifier hairpins were purchased from Molecular Instruments. Imaging was carried out on a Zeiss LSM980 Airyscan 2 microscope, with 405 nm, 488 nm, 561 nm and 639 nm laser filter sets, and using a 10×/0.45 lens. Z-stack images were analysed using Fiji/ImageJ (Guiet et al., 2021; Schindelin et al., 2012).

### Bulk RNA sequencing and GO term analysis
RNA was extracted from pools of 48 gastruloids developed with either HM-N2B27 or NDiff227 using the RNeasy Micro Kit (Qiagen; 74004), and its concentration and purity were determined using PicoGreen and Fragment Analyzer (Agilent). A total of three independent experimental/ biological replicates were made for each condition. Library preparation was carried out by the CRG Genomics Facility (Spain), using the TruSeq stranded mRNA Library Prep (ref. 20020595, Illumina) according to the manufacturer's protocol. Sequencing was performed, also at the CRG Genomics Facility, in NextSeq 2000 and generated around 30 M paired-end reads per sample. The preprocessing and downstream analysis of the bulk RNAseq data was carried out in accordance with Dias et al. (2025b). For the GO term analysis, genes with an adjusted $P$-value<0.05, determined via the Benjamini-Hochberg method for multiple testing correction, and an absolute $log_2$ fold change >1 were classified as significantly differentially expressed (DEGs). Over-representation analysis (ORA) of the DEGs was conducted using enrichR (version 3.2) and validated using WebGestalt 2024 (Table S5). The custom background gene list used for ORA was chosen to include all genes with non-zero counts from the bulk RNAseq analysis (Table S5).

### Acknowledgements
We thank Jenny Nichols, Carla Mulas and Filomena Amoroso for advice and discussions regarding N2B27, Anna Bigas for the E14Tg2A cell line, Gordon Keller for the Bra::GFP cell line (Fehling et al., 2003), Anna-Katerina Hadjantonakis for the H2B::mCherry;GFP-GFP cell line (Nowotschin et al., 2009) and Vikas Trivedi for the *Tfap2c* HCR probe. We are indebted to the University of Liverpool's Centre for Cell Imaging (CCI) facility for technical support and provision of state-of-the-art imaging equipment funded by the Biotechnology and Biological Sciences Research Council (BB/R01390X/1) and to the CRG Genomics and Imaging facilities at the PRBB (Barcelona, Spain).

### Competing interests
A.M.A. is an inventor in two patents on Human Polarised Three-dimensional Cellular Aggregates PCT/GB2019/052670 and Polarised Three-dimensional Cellular Aggregates PCT/GB2019/052668.

### Author contributions
Conceptualization: A.M.A., A.D., D.A.T.; Methodology: T.B., A.D., D.A.T.; Formal analysis: S.L., P.P.-M., A.V., A.D., D.A.T.; Investigation: T.B., U.-M.F., J.D.F., J.B.J.G., S.L., H.Y.G., R.F.P., A.D.; Writing – original draft: A.D., D.A.T.; Writing – review & editing: T.B., S.L., P.P.-M., U.-M.F., A.V., J.D.F., J.B.J.G., H.Y.G., R.F.P., A.M.A., A.D., D.A.T.; Supervision: A.M.A., A.D., D.A.T.; Funding acquisition: J.D.F., A.M.A., A.D., D.A.T.

### Funding
D.A.T. was funded in this work by the Biotechnology and Biological Sciences Research Council through a New Investigator Grant (BB/X000907/1) and a Strategic Longer and Larger (sLoLa) Programme Grant (BB/Y00311X/1), and by the National Centre for the Replacement, Refinement and Reduction of Animals in Research (NC/T002131/1). A.M.A. was funded by an European Research Council Advanced Grant (MiniEmbryoBlueprint 834580) and by the 'Maria de Maeztu' Program for Units of Excellence in R&D of the Ministerio de Ciencia, Innovación y Universidades (CEX2018-000792-M). J.D.F. was funded by a 'la Caixa' Foundation (100010434) junior leader fellowship project (LCF/BQ/PI23/11970017). A.D. and J.D.F. were funded by European Molecular Biology Organization Postdoctoral Fellowships (ALTF 948-2022 to A.D.; ALTF 605-2022 to J.D.F.). Open Access funding provided by the University of Liverpool. Deposited in PMC for immediate release.

### Data and resource availability
Raw and processed RNA sequencing data have been deposited in Biostudies/ ArrayExpress under the accession number E-MTAB-14893. The bioinformatic analysis pipeline, including all code and parameters, is available in GitHub: https://github.com/stembryo-lab/HM-N2B27_vs_NDiff227_gastruloids and https://github.com/gastruloids/N2B27_vs_NDiff. All other relevant data and details of resources can be found within the article and its supplementary information.

### The people behind the papers
This article has an associated 'The people behind the papers' interview with some of the authors.

### Peer review history
The peer review history is available online at https://journals.biologists.com/dev/ lookup/doi/10.1242/dev.204774.reviewer-comments.pdf

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
