## [Peer Review File · Development (Cambridge, England)]

N2B27 media formulations influence gastruloid development

Tina Balayo, Sharna Lunn, Pau Pascual-Mas, Ulla-Maj Fiuza, Amruta Vasudevan, Joshua D. Frenster, Joel B. Josende García, Hannah Y. Galloon, Raquel Flores Peirats, Alfonso Martinez Arias, André Dias and David A. Turner

DOI: 10.1242/dev.204774

Editor: Matthias Lutolf

Review timeline

Original submission:	7 March 2025
Editorial decision:	15 April 2025
First revision received:	13 August 2025
Accepted:	2 October 2025

Original submission

First decision letter

MS ID#: dev.204774

MS Title: N2B27 media formulations influence gastruloid development

Authors: Tina Balayo; Sharna Lunn; Pau Pascual-Mas; Ulla-Maj Fiuza; Amruta Vasudevan; Joshua D. Frenster; Joel B. Josende García; Hannah Y. Galloon; Raquel Flores Peirats; Alfonso Martinez Arias; André Dias; David A. Turner

Article Type: Research Report

Dear Dr Turner,

I have now received all the referees' reports on the above manuscript, and have reached a decision. The referees' comments are appended below, or you can access them online: please go to:

As you will see, the referees express considerable interest in your work, but have some significant criticisms and recommend a substantial revision of your manuscript before we can consider publication. If you are able to revise the manuscript along the lines suggested, which may involve further experiments, I will be happy receive a revised version of the manuscript. Your revised paper will be re-reviewed by one or more of the original referees, and acceptance of your manuscript will depend on your addressing satisfactorily the reviewers' major concerns. Please also note that Development will normally permit only one round of major revision. If it would be helpful, you are welcome to contact us to discuss your revision in greater detail. Please send us a point-by-point response indicating your plans for addressing the referees' comments, and we will look over this and provide further guidance.

Please attend to all of the reviewers' comments and ensure that you clearly highlight all changes made in the revised manuscript. Please avoid using 'Tracked changes' in Word files as these are lost in PDF conversion. I should be grateful if you would also provide a point-by-point response detailing how you have dealt with the points raised by the reviewers in the 'Response to Reviewers' box. If you do not agree with any of their criticisms or suggestions please explain clearly why this is so.

Reviewer 1

SUMMARY OF THE ADVANCE MADE IN THIS PAPER AND ITS POTENTIAL SIGNIFICANCE TO THE FIELD

In recent years, in vitro embryo models have seen widespread adoption within the developmental biology and biosystems communities. One of these models is gastruloid, a system developed in the laboratory of Alfonso Martinez Arias, which mimic many developmental events that take place early during the gastrulation of the embryo. Like in all in vitro systems, gastruloid culture relies on a strict protocol involving precise steps, complex culture media and signalling molecules sourced from diverse suppliers. Within the gastruloid community, factors that introduce variations in gastruloid size and cellular composition have been a topic of ongoing debate. For example, batch-to-batch variability and discrepancies observed when using different cell lines remain unresolved.

In this study, Balayo and colleagues demonstrate consistent differences in mouse gastruloid development when using home-made N2B27 medium (HM-N2B27) versus its commercially available counterpart, the NDiff227. The work focuses on evaluating morphological changes (e.g., size and shape) and alterations in gene expression patterns in gastruloids cultured under these two conditions. While descriptive in nature, the study provides clear evidence supporting its central hypothesis: that the choice of culture medium significantly influences key parameters of gastruloid development.

I believe the current work will be of interest not only to researchers considering adopting gastruloids in their studies -a growing number of laboratories- but also to those already using gastruloids who face issues in their culture systems. The data appear robust, and the text is well-written. However, one key issue is the lack of explanation for the differences observed by the authors when comparing the two culture media. While the authors thoroughly acknowledge the limitations of their study (a fair and transparent approach), this does not preclude the need for some recommendations. Providing such guidance would enhance, in my opinion, the practical value of the work.

Nevertheless, I recommend to consider this paper for publication in Development due to its novelty to addressing the variability question in these in vitro systems, a challenge that will remain central to the field.

SUGGESTIONS TO AUTHORS

I. Data description and representation

Due to the nature of the study, and since this is the main goal of the paper, the authors should clarify whether the term 'replicates' refers to technical or biological replicates. How were the replicates generated? Individual gastruloids or pooled? The authors must also clearly state the number of gastruloids or plates quantified for each figure legend.

For example, line 99 "(>90%; n = 5 replicates; > 100 individual gastruloids; Fig. 1A & B)". Are these 100 individual gastruloids that were quantified over 5 replicates?

Line 326, "...Three independent experiments were done to generate three 96-well plates of E14-Tg2A gastruloids made with either HM-N2B27 or NDiff227 (48 gastruloids per sample, per plate)." Again, it's not clear what the number n is, and if it's n = 3, what each n represents.

The authors mentioned that the study was conducted "in two laboratories". Which experiment was done in which lab and why did the author think it was important to mention this?

II. Data interpretation

The authors presented a bulk RNA-seq dataset to show the variation in cell identity between the two conditions. Gastruloid in HM-N2B27 produced more spinal cord-related genes, while the NDiff227 condition produced a majority mesodermal lineage. This neural/mesodermal bias has already been described (e.g <https://www.biorxiv.org/content/10.1101/2022.09.27.509783v1.full>). How can the authors exclude the possibility that the variation observed with the two media is not the consequence of the batch effect that this system can produce? For example, if the authors compare their datasets to the publicly available RNA-seq data from gastruloid culture, will the clear separate clustering between the two conditions still be visible?

Another question regarding the RNA-seq and its potential link to the number of cells in gastruloids, which was clearly reduced in the NDiff227 condition. Could the authors comment on whether the

variation in cell identity between the two conditions could be related to the number of cells, especially in light of the results of this paper (<https://www.biorxiv.org/content/10.1101/2024.12.23.630037v1.full>).

Reviewer 2

SUMMARY OF THE ADVANCE MADE IN THIS PAPER AND ITS POTENTIAL SIGNIFICANCE TO THE FIELD

In this manuscript, Balayo et. al. report an important comparative study investigating the effects of commercially available NDiff227 and home made N2B27 on gastruloid formation. They quantify morphological features and conclude that HM-N2B27 yields to formation of bigger and more reproducible gastruloids. They perform bulk RNA sequencing and discover that HM-N2B27 gastruloids contain more of the neuroectoderm tissues and NDiff227 gastruloids contain more mesoderm derivatives.

This paper challenges long standing culture medium standards in the gastruloid field and will help researchers reach a consensus on a standard mouse gastruloid N2B27 formulation.

SUGGESTIONS TO AUTHORS

General comments:

In the manuscript, successful gastruloid formation is largely quantified based on morphological parameters (length, area and elongation index). It is crucial to validate and compare the tissue organization in HM-N2B27 and NDiff227 gastruloids. Immunostainings showing spatial arrangement of mesoderm, endoderm and ectoderm derivatives should be provided.

In line 129-130, the authors comment that more frequent multi-axis formation in HM-N2B27 could be due to the increased number of cells in this media. They should support this claim by performing an experiment starting with a lower number of cells and quantifying the frequency of multi-axis formation in HM-N2B27.

Why do the HM-N2B27 gastruloids have more neural tissues? Can it be due to the presence of vitamin A in the B27 supplement that can be converted into retinoic acid (RA)? Do NDiff227 also contain vitamin A?

There are reports on the role of RA in gastruloid elongation (Hamazaki et. al., Nature Cell Biology, 2024). If NDiff227 contains vitamin A, could its degradation during media transport or thawing be the reason behind limited neural differentiation and irreproducible gastruloid elongation?

The authors report that following 48h, HM-N2B27 gastruloids get bigger than NDiff227. This is the time point when RA signaling was shown to be crucial for gastruloid development (Hennesy et. al., biorxiv, 2023). The authors can supplement RA to NDiff227 at 48h to see whether this improves neural differentiation and gastruloid elongation.

HM-N2B27 recipes prepared in Liverpool and Barcelona contain L-glutamine or Glutamax, respectively. L-glutamine is temperature and pH sensitive and therefore less stable in culture. The authors should directly compare gastruloids formed from both media to eliminate glutamine-based effects.

At 72h AA, gastruloids are washed two times with 150ul N2B27? This is different from the original protocols cited in the paper. The authors should demonstrate the effect of double washing on gastruloid formation.

Immunostaining for Cdh1 shows less expression in NDiff227 at 96h, however, bulk RNAseq shows the more Cdh1 expression in NDiff227 compared to HM-N2B27 at 120h. The authors should comment on this.

It would be interesting to stain for the PGCs as the bulk RNAseq cannot differentiate between PGC and pluripotency signatures.

What is the reason behind using three different mESC maintenance media (10% serum, 15% serum, 2iL) and forming gastruloids in different U-bottom ULA plates? In a study that aims to compare different formulations of N2B27, all the other parameters should be kept constant.

Do all tested batches of commercially available N2 and B27 supplements lead to reproducible gastruloid elongation?

Specific remarks:

In figure 1 and 2, it is not stated which E14 cells were used. Are they the 2iL precultured ones or ESL (10% or 15%) ones?

Supplementary Fig. 2: It is impossible to read the data. Higher resolution graphs are necessary.

Supplementary Tables 3-5 are missing.

Reviewer 3

The report by Balayo et al. deals with an important challenge in the stem-cell-based embryo model and organoid field in general: (how) does the culture medium influence the differentiation outcome? In particular, they study the effect of home-made vs commercial N2B27 on gastruloid differentiation outcome. Although there's plenty of anecdotal evidence that the use of home-made vs commercial N2B27 affects the outcome, this has not been studied systematically. This is exactly what Balayo et al. set out to do, and they should be applauded for that. The work is timely, important, and highly relevant to the field. Studies like these are critically needed to ensure that the field can make solid progress on the long term. While I'm therefore very supportive of the work, there are some critical issues (see below). Once these are addressed appropriately, I would highly recommend publication in Development.

Major points:

1. A major concern is that the authors report "both media enable the standard gastruloid elongation" (Abstract). However, in the Methods section, they state that "Gastruloids that did not elongate were removed from the analysis". Importantly, the % not elongating is different for commercial vs homemade N2B27. The authors should clearly report this difference in elongation capacity in the main text and Figure 1, and adjust their conclusions accordingly.

2. The authors report that the findings were replicated in two labs, which is very important. However, the quantifications do not show which data were acquired in which lab. This would be useful to assess lab-to-lab variability (and how this relates to inter-experimental variability). Moreover, it seems that both labs did not use the exact same recipe for homemade N2B27. One of the DMEM/F12 is without glutamine, the other with. In both cases, the authors add L-Glutamine or Glutamax. Hence, there are, potentially important, differences in medium conditions. Did the authors test if this affected outcome?

3. The authors report that both types of N2B27 homemade media were batch-tested to assess their ability to generate gastruloids and the degree of morphological inter- and intra-batch variation". When did they in/exclude media batches for the analysis? It is important to report quantitative criteria for this.

4. Figure 2 is an interesting finding, but lacks quantifications and statistics.

5. In the limitations, the authors state that single-cell RNA-sequencing is "unlikely to provide substantial additional insights". I strongly disagree. Two preprints (Rosen et al., 2022 & Villaronga Luque et al., 2023) have clearly demonstrated the value of scRNA-seq to assess gastruloid variability beyond differential gene expression based on bulk measurements. Such analysis would e.g. allow the authors to test if higher expression of a certain gene is the result of higher expression in all cells, or due to a difference in cellular proportions. It would also allow the authors to look at cell type specific differences in gene expression. That is not to say the authors should conduct scRNA-seq for the revisions. I agree it is costly, and, as it is, the bulk-seq data suffice to

support the conclusions of the study. But they should remove the statement that scRNA-seq would not provide substantial additional insights, since it's simply not true.

6. The authors tested the effects of homemade vs commercial N2B27 in multiple different cell lines cultured under different pluripotency conditions (S1) but do not provide quantifications of the differences. They should show the quantifications that demonstrate that there are "consistent phenotypic differences across cell lines, culture conditions, and laboratories" as they claim. Related: The Bra::GFP line that is used (Fehling et al, Development 2003) is also derived from WT ESCs with an E14 background. Given that the other cell lines used are also E14, the authors should make clear in the limitations that it remains unclear how well the findings can be extrapolated to gastruloids derived from mESCs with a different genetic background.

Minor points:

1. There are a few details regarding Gastruloid generation that need clarification / justification since these steps deviate from conventional gastruloid culture protocols:

a. ULA plates were used only for cells pre-cultured with higher levels of serum. Why? And why were different serum conditions used in first place? How does this affect the outcome variables? How does the use of ULA vs non-ULA plates affect outcome variables?

b. The outer wells of the 96-well plates were not used. What is the reason for that? How would using the outer wells affect the conclusions?

c. The reported number of cells seeded per well was 200-300 cells. There have been multiple reports showing how gastruloid size affects their developmental dynamics (Bennabi et al., BioRxiv 2024; Fiuza et al., BioRxiv 2024; van den Brink et al., Development 2014; Anlas et al., Development 2024).

d. Why are the gastruloids washed twice at 72h AA?

2. Line 61: for initial number of cells, the authors should also cite Anlas et al., Development 2024 (PMID: 39552366).

3. Line 40-41: "cell fate decisions... coordinate cell movements and tissue morphogenesis" - this somewhat overstates the importance of cell fate decisions (relative to e.g. physical properties of the system).

4. Line 58: what do the authors mean by "maintain their reproducibility"?

5. Line 62-63: what do the authors mean by "precision in the execution of the protocol"? Isn't precise execution of the protocol simply a manner of good scientific practice?

6. Line 111-112: "NDiff227 gastruloids display a more consistent shape from pole to pole". What do the authors mean by "more consistent shape"? What are the metrics measured to quantify this?

7. It is unclear to me what the exact relevance of using MEMs is in this context. The authors should better clarify this. To understand the analytical framework & assess its relevance and importance, it would be important to provide access to the Lunn et al. manuscript mentioned as "in preparation". I strongly recommend that the authors cite this as a preprint in the revised version of this manuscript.

8. Line 132-134: I don't understand the implied link between how the medium choice affects mentioned cell behaviors and cell competition. Do the authors suggest that there is less apoptosis in homemade N2B27?

9. Line 152: this should be $p < 0.05$ as stated in Methods? Also, the Methods mention a logFC cut-off of 1, whereas the main text mentions a logFC cut-off of 1.5

10. Line 154: what is meant by "significant distance". I don't see a statistical analysis?

11. Line 195-197: could the authors distinguish between PGC-like cells & ectopic pluripotent cells?

12. Line 204: since Hedgehog signalling is enriched in HM-N2B27 gastruloids, do the authors observe more endoderm and/or axial mesoderm?

13. Line 229:230: if the authors really want to make the point that the rate of cell proliferation, survival and/or differentiation is different, they should conduct experiments that show this (e.g. pH3, cleaved caspase stains).

14. Figure S3: distinct morphology needs to be quantified.

15. For some metrics, e.g. elongation index, it seems that while the median is similar, the variance is different between conditions. The authors should comment on this.

First revision

Author response to reviewers' comments

Reviewer 1:

SUMMARY OF THE ADVANCE MADE IN THIS PAPER AND ITS POTENTIAL SIGNIFICANCE TO THE FIELD: *In recent years, in vitro embryo models have seen widespread adoption within the developmental biology and biosystems communities. One of these models is gastruloid, a system developed in the laboratory of Alfonso Martinez Arias, which mimic many developmental events that take places early during the gastrulation of the embryo. Like in all in vitro systems, gastruloid culture relies on a strict protocol involving precise steps, complex culture media and signalling molecules sourced from diverse suppliers. Within the gastruloid community, factors that introduce variations in gastruloid size and cellular composition have been a topic of ongoing debate. For example, batch-to- batch variability and discrepancies observed when using different cell lines remain unresolved.*

In this study, Balayo and colleagues demonstrate consistent differences in mouse gastruloid development when using home-made N2B27 medium (HM-N2B27) versus its commercially available counterpart, the NDiff227. The work focuses on evaluating morphological changes (e.g., size and shape) and alterations in gene expression patterns in gastruloids cultured under these two conditions. While descriptive in nature, the study provides clear evidence supporting its central hypothesis: that the choice of culture medium significantly influences key parameters of gastruloid development.

I believe the current work will be of interest not only to researchers considering adopting gastruloids in their studies -a growing number of laboratories- but also to those already using gastruloids who face issues in their culture systems. The data appear robust, and the text is well-written. However, one key issue is the lack of explanation for the differences observed by the authors when comparing the two culture media. While the authors thoroughly acknowledge the limitations of their study (a fair and transparent approach), this does not preclude the need for some recommendations. Providing such guidance would enhance, in my opinion, the practical value of the work.

Nevertheless, I recommend to consider this paper for publication in Development due to its novelty to addressing the variability question in these in vitro systems, a challenge that will remain central to the field.

We would like to thank Reviewer 1 for reviewing our manuscript and for the favourable comments, particularly the recommendation for it to be published in *Development*.

SUGGESTIONS TO AUTHORS

1. Data description and representation

- a. *Due to the nature of the study, and since this is the main goal of the paper, the authors should clarify whether the term 'replicates' refers to technical or*

biological replicates. How were the replicates generated? Individual gastruloids or pooled? The authors must also clearly state the number of gastruloids or plates quantified for each figure legend. For example, line 99 "(>90%; n = 5 replicates; > 100 individual gastruloids; Fig. 1A & B)". Are these 100 individual gastruloids that were quantified over 5 replicates?

We have now addressed these points raised by the reviewer in the new version of our manuscript. The term "replicate" was mostly used to refer to biological replicates. We agree that it is important to clarify these issues and thank the reviewer for highlighting it.

b. Line 326, "...Three independent experiments were done to generate three 96-well plates of E14Tg2A gastruloids made with either HM-N2B27 or NDiff227 (48 gastruloids per sample, per plate)." Again, it's not clear what the number n is, and if it's n = 3, what each n represents.

We modified this part - now reads: "RNA was extracted from pools of 48 gastruloids, developed with either HM- N2B27 or NDiff227, using the Qiagen RNeasy Micro Kit (74004) and its concentration and purity were determined using PicoGreen and Fragment Analyzer (Agilent). A total of three independent experimental/biological replicates were made per condition".

c. The authors mentioned that the study was conducted "in two laboratories". Which experiment was done in which lab and why did the author think it was important to mention this?

We removed this comment but have indicated in the methods which experiments were performed in which lab.

2. Data interpretation: The authors presented a bulk RNA-seq dataset to show the variation in cell identity between the two conditions. Gastruloid in HM-N2B27 produced more spinal cord-related genes, while the NDiff227 condition produced a majority mesodermal lineage. This neural/mesodermal bias has already been described (e.g. <https://www.biorxiv.org/content/10.1101/2022.09.27.509783v1.full>). How can the authors exclude the possibility that the variation observed with the two media is not the consequence of the batch effect that this system can produce? For example, if the authors compare their datasets to the publicly available RNA-seq data from gastruloid culture, will the clear separate clustering between the two conditions still be visible?

The batch effect possibility is indeed an important consideration and that is why the experiment was independently repeated three times. Given the high consistency observed across these experiments, we don't think that the reported batch effect could explain the differences we noticed between gastruloids developed using homemade or commercial N2B27.

As for expanding the analysis by including publicly available RNA-seq data, we believe this is not a good approach because the differences we are looking at are relatively small and this level of high detail can be obscured by potential variability introduced through the use of different cell lines, 2D culturing conditions (see for example Blotenburg et al., 2025) and/or methods used to undertake the RNAseq.

In any case, to further highlight the robustness of the homemade media (questioned by Reviewer 2) and the neural-mesodermal variation, please find below a PCA including the bulk RNAseq samples described in the paper plus two more HM-N2B27 samples ("HM2-N2B27"), processed at the same time as the previous, but made using distinct batches of N2 and B27 (**left**). These new samples cluster with their homemade counterparts and the differences against the commercial N2B27 ("C-N2B27") samples are maintained, as indicated by the top 100 genes from the loadings of PC 1 and 2 (**right**).

3. *Another question regarding the RNA-seq and its potential link to the number of cells in gastruloids, which was clearly reduced in the NDiff227 condition. Could the authors comment on whether the variation in cell identity between the two conditions could be related to the number of cells, especially in light of the results of this paper (<https://www.biorxiv.org/content/10.1101/2024.12.23.630037v1.full>).*

Yes, it is possible that the observed variation in neural and mesodermal-associated gene expression in the two types of gastruloids can be partially caused by differences in the number of cells. However, we don't see a strong argument for it. Bennabi et al., 2024 and Fiuza et al., 2024, independently observed that gastruloids developed from a very small number of cells display increased neural-related gene expression at 120h, while gastruloids formed with a higher number of cells preferentially express mesodermal-like genes. In our work we noticed the opposite trend: HM-N2B27 gastruloids exhibited both a higher number of cells and increased neural-associated gene expression compared to NDiff227 gastruloids which, despite being smaller, display higher mesodermal gene expression. Furthermore, reducing the number of cells in the homemade condition (from 300 to 200 cells) did produce any major effect in terms of gene expression for the genes probed via HCR. This result is in line with observations from both Bennabi et al. and Fiuza et al., suggesting that transcriptional programs and cell fate composition in mouse gastruloids are stable within a size range.

Reviewer 2:

SUMMARY OF THE ADVANCE MADE IN THIS PAPER AND ITS POTENTIAL SIGNIFICANCE TO THE FIELD: In this manuscript, Balayo et. al. report an important comparative study investigating the effects of commercially available NDiff227 and home-made N2B27 on gastruloid formation. They quantify morphological features and conclude that HM-N2B27 yields to formation of bigger and more reproducible gastruloids. They perform bulk RNA sequencing and discover that HM-N2B27 gastruloids contain more of the neuroectoderm tissues and NDiff227 gastruloids contain more mesoderm derivatives.

This paper challenges long standing culture medium standards in the gastruloid field and will help researchers reach a consensus on a standard mouse gastruloid N2B27 formulation.

We want to thank Reviewer 2 for reviewing our manuscript and for the favourable comments, especially regarding the importance of our work.

SUGGESTIONS TO AUTHORS

General comments:

- In the manuscript, successful gastruloid formation is largely quantified based on morphological parameters (length, area and elongation index). It is crucial to validate and compare the tissue organization in HM-N2B27 and NDiff227 gastruloids.*

Immunostainings showing spatial arrangement of mesoderm, endoderm and ectoderm derivatives should be provided.

We have performed HCR for key marker genes in the two types of gastruloids, and the results are displayed in Fig. 3.

- In line 129-130, the authors comment that more frequent multi-axis formation in HM-N2B27 could be due to the increased number of cells in this media. They should support this claim by performing an experiment starting with a lower number of cells and quantifying the frequency of multi-axis formation in HM-N2B27.*

We have performed these experiments and included them in the new version of the manuscript (**Supplemental Fig. S1F**). The results are in agreement with our initial suggestion.

- Why do the HM-N2B27 gastruloids have more neural tissues? Can it be due to the presence of vitamin A in the B27 supplement that can be converted into retinoic acid (RA)? Do NDiff227 also contain vitamin A? There are reports on the role of RA in gastruloid elongation (Hamazaki et. al., Nature Cell Biology, 2024). If NDiff227 contains vitamin A, could its degradation during media transport or thawing be the reason behind limited neural differentiation and irreproducible gastruloid elongation? The authors report that following 48h, HM- N2B27 gastruloids get bigger than NDiff227. This is the time point when RA signaling was shown to be crucial for gastruloid development (Hennesy et. al., biorxiv, 2023). The authors can supplement RA to NDiff227 at 48h to see whether this improves neural differentiation and gastruloid elongation.*

We agree that vitamin A (and by extension, retinoic acid) plays an important role in gastruloid development and neural differentiation, and that it is likely present in NDiff227. However, since the composition of this commercial medium is proprietary, we cannot confirm this with certainty.

We agree that the hypothesis regarding potential vitamin A degradation is plausible, but we believe that any experiment aiming to test this would face important limitations. In addition to the uncertainty regarding the presence of vitamin A in the media in NDiff227, the degradation *per se* would be an experimental variable that we cannot measure and that would likely vary between and within batches. This variability will likely have a significant impact on reproducibility and, therefore, preclude us to draw any conclusions from such experiments.

- HM-N2B27 recipes prepared in Liverpool and Barcelona contain L-glutamine or Glutamax, respectively. L- glutamine is temperature and pH sensitive and therefore less stable in culture. The authors should directly compare gastruloids formed from both media to eliminate glutamine-based effects.*

The reviewer is quite right to point out the issues regarding the stability of L-glutamine compared with the more stable Glutamax, however this is not going to impact our conclusions, nor indeed necessitate further comparisons for the following reasons:

1. Firstly, L-Glutamine was kept at -70°C and taken to 4°C only the day before the preparation of HM-N2B27.
2. The N2B27 from both labs were made fresh, aliquoted, stored appropriately at 4°C, and disposed of after two weeks even if bottles had not been used (as mentioned in the materials and methods); this ensures temperature stability, and also pH stability (maintained at ~7).
3. When diluted in media, L-glutamine shows no appreciable degradation (0.1% per day) in these conditions over the duration we keep the media; see these papers Kahn and Elia (DOI:10.1016/0261-5614(91)90037-D) and Maja Jagušić *et al.* (DOI:10.1007/s10616-015-9875-8). L-glutamine's spontaneous degradation would not be significantly different to Glutamax in our storage conditions.
4. We do however use N2B27 at 37°C which as the reviewer is aware, will have a more pronounced effect on the stability of L-glutamine. However, we mitigate this by changing the media each day, and over this 24h timeframe, degradation of L-

glutamine is negligible and comparable to Glutamax. This is illustrated in Figure 3 in Thermo's product page for Glutamax (see this hyperlink) when the initial concentration is normalised.

Conscious that as this manuscript is exacting in the detail regarding the effects of media conditions on gastruloid development, we have decided to add a clarifying statement in the materials and methods: "*The storage conditions are critical especially if L-glutamine is used instead of Glutamax, as improper storage will lead to its degradation (Jagusic et al., 2016; Mulas et al., 2019).*"

5. *At 72h AA, gastruloids are washed two times with 150ul N2B27? This is different from the original protocols cited in the paper. The authors should demonstrate the effect of double washing on gastruloid formation.*

We agree and have removed the data obtained with two N2B27 washes at 72h (Fig. S1A, E14Tg2A gastruloids - 2iL preculture). New data is provided now following the standard protocol.

6. *Immunostaining for Cdh1 shows less expression in NDiff227 at 96h, however, bulk RNAseq shows the more Cdh1 expression in NDiff227 compared to HM-N2B27 at 120h. The authors should comment on this.*

In our opinion the variation observed for Cdh1 at a protein level between 90 and 96h should not be linked to the transcriptomic differences noticed at 120h between NDiff227 and HM-N2B27 conditions. The reason for this is that the protein staining at day 4 reflects the emergence of the caudal epiblast and the neuromesodermal cell population, whereas the higher Cdh1 expression levels at 120h are likely related to the increased amount of ectopic pluripotency/PGCs in NDiff227 gastruloids.

7. *It would be interesting to stain for the PGCs as the bulk RNAseq cannot differentiate between PGC and pluripotency signatures.*

As highlighted in Cooke et al. 2023, gastruloid-derived PGC-like cells co-express pluripotency and PGC-associated markers thus it is difficult to make a separation between the two signatures at 120h. We did HCR for *Tfap2c*, a transcription factor that plays a critical role in the development of PGCs and observed that it is expressed in both types of gastruloids. A similar pattern was observed for *Oct4*, which is more associated with pluripotency (Fig. 3).

8. *What is the reason behind using three different mESC maintenance media (10% serum, 15% serum, 2iL) and forming gastruloids in different U-bottom ULA plates? In a study that aims to compare different formulations of N2B27, all the other parameters should be kept constant.*

The main work (Figures 1 - 4), was developed using cells cultured in the same conditions (10% ESLIF), and the N2B27 media was the only variable in these experiments. The experiments displayed in Figure S1 aimed to understand whether there were also differences between gastruloids formed in the two N2B27 media conditions if: 1) the cells were developed in other culture conditions, such as the use of 2i; 2) the aggregation was facilitated through the use of ULA plates and 3) the cells were from a non E14Tg2A background. Importantly, all these conditions (which span those generally used by the gastruloid community) were tested separately, and the conclusions regarding the phenotypes between NDiff227 and HM-N2B27 are constant - we have made this point clear in the new version of the manuscript.

9. *Do all tested batches of commercially available N2 and B27 supplements lead to reproducible gastruloid elongation?*

Yes.

Specific remarks:

1. *In figure 1 and 2, it is not stated which E14 cells were used. Are they the 2iL precultured ones or ESL (10% or 15%) ones?*

All the work was carried out using E14Tg2A cells pre-cultured in 10% ESL, whereas stated otherwise. We have made this point now clearer in both the figures and the main text.

2. *Supplementary Fig. 2: It is impossible to read the data. Higher resolution graphs are necessary.*

We apologise for this oversight of figure resolution. To make it clearer, we have now included three full-sized figures which have been relabelled individually as Fig. S2A, Fig. S2B, and Fig. S2C. We think it is best that these still have the “Fig. 2” appellation, but to give each figure maximum resolution, they are provided separately for ease of reading.

3. *Supplementary Tables 3-5 are missing.*

We apologise to the reviewer for this as we thought this was included. The main manuscript file does not have these tables included (even though they’re labelled as being there), but we assumed they were uploaded as separate entities. We will ensure these tables are uploaded for the revision.

Reviewer 3:

The report by Balayo et al. deals with an important challenge in the stem-cell-based embryo model and organoid field in general: (how) does the culture medium influence the differentiation outcome? In particular, they study the effect of home-made vs commercial N2B27 on gastruloid differentiation outcome. Although there's plenty of anecdotal evidence that the use of home-made vs commercial N2B27 affects the outcome, this has not been studied systematically. This is exactly what Balayo et al. set out to do, and they should be applauded for that. The work is timely, important, and highly relevant to the field. Studies like these are critically needed to ensure that the field can make solid progress on the long term. While I'm therefore very supportive of the work, there are some critical issues (see below). Once these are addressed appropriately, I would highly recommend publication in Development.

We would like to thank Reviewer 3 for reviewing our manuscript and for all the favourable comments, especially the recommendation for it to be published in *Development*.

Major points:

1. *A major concern is that the authors report "both media enable the standard gastruloid elongation" (Abstract). However, in the Methods section, they state that "Gastruloids that did not elongate were removed from the analysis". Importantly, the % not elongating is different for commercial vs homemade N2B27. The authors should clearly report this difference in elongation capacity in the main text and Figure 1, and adjust their conclusions accordingly.*

The difference in elongation capacity is now reported in both the main text and Fig.1, as suggested by the reviewer.

2. *The authors report that the findings were replicated in two labs, which is very important. However, the quantifications do not show which data were acquired in which lab. This would be useful to assess lab-to-lab variability (and how this relates to inter-experimental variability).*

We mentioned “we undertook a short study across two laboratories”. However, to avoid any possible misunderstanding, we removed this comment because not all the findings were replicated in the two labs (e.g. transcriptomic experiments).

- a. *Moreover, it seems that both labs did not use the exact same recipe for homemade N2B27. One of the DMEM/F12 is without glutamine, the other with. In both cases, the authors add L-Glutamine or Glutamax. Hence, there are, potentially important, differences in medium conditions. Did the authors test if this affected outcome?*

This comment is similar to a question from Reviewer 2, and we would like to refer this reviewer to the answer we gave previously regarding L-glutamine vs Glutamax in our culture conditions, and the subsequent modification in the manuscript.

3. *The authors report that both types of N2B27 homemade media were batch-tested to assess their ability to generate gastruloids and the degree of morphological inter- and intra-batch variation". When did they in/exclude media batches for the analysis? It is*

important to report quantitative criteria for this.

All new N2B27 media batches were tested against the ones in use, and these were discarded if the amount of non-elongated gastruloids was higher than the reported percentages (10% for homemade N2B27 and 20% for NDiff227) or in case of significant morphological abnormalities. Although all tested homemade N2B27 batches were fine, we found that some commercial batches were not suitable for gastruloid development (e.g. Lot AM60017S).

4. Figure 2 is an interesting finding but lacks quantifications and statistics.

To do a proper quantification and statistics of the results shown in Figure 2, we performed new staining experiments and used light-sheet microscopy to enable the quantification of more NDiff227 and HM-N2B27 gastruloids at both 90 and 96h. These images, we believe, are much clearer. We have quantified the expression pattern of Cdh2, and that is now shown in Fig. 2F). We have also modified this figure to bring in data we had reserved for the supplemental material, and we have now included statistical tests showing the difference in roundness from the different media at 96h.

The new quantification robustly supports the statements we previously made in the manuscript where we show how gastruloids grown in commercial NDiff227 are developmentally delayed compared with HM-N2B27: the frequency of gastruloids initiating morphogenetic changes (ovoid/protrusion phenotype) is greater in HM-N2B27, reflected by a lower roundness value (e.g. no longer spherical), and the delayed expression of Cdh2.

5. In the limitations, the authors state that single-cell RNA-sequencing is "unlikely to provide substantial additional insights". I strongly disagree. Two preprints (Rosen et al., 2022 & Villaronga Luque et al., 2023) have clearly demonstrated the value of scRNA-seq to assess gastruloid variability beyond differential gene expression based on bulk measurements. Such analysis would e.g. allow the authors to test if higher expression of a certain gene is the result of higher expression in all cells, or due to a difference in cellular proportions. It would also allow the authors to look at cell type specific differences in gene expression. That is not to say the authors should conduct scRNA-seq for the revisions. I agree it is costly, and, as it is, the bulk-seq data suffice to support the conclusions of the study. But they should remove the statement that scRNA-seq would not provide substantial additional insights, since it's simply not true.

We have removed this statement.

6. The authors tested the effects of homemade vs commercial N2B27 in multiple different cell lines cultured under different pluripotency conditions (S1) but do not provide quantifications of the differences. They should show the quantifications that demonstrate that there are "consistent phenotypic differences across cell lines, culture conditions, and laboratories" as they claim. Related: The Bra::GFP line that is used (Fehling et al, Development 2003) is also derived from WT ESCs with an E14 background. Given that the other cell lines used are also E14, the authors should make clear in the limitations that it remains unclear how well the findings can be extrapolated to gastruloids derived from mESCs with a different genetic background.

The presence of a scale bar enables the reader to compare the size and elongation of the two types of gastruloids. Although the behaviour of the Bra::GFP cells is significantly different from that of standard WT E14Tg2A cells (e.g. Stapornwongkul et al., 2025), we performed new experiments using a cell line from an R1 genetic background (H2B-mCherry;GPI-GFP). These experiments supported our original claim, as we observed consistent differences in gastruloid development when cultured in NDiff227 versus HM-N2B27 (Fig. S1C).

Minor points:

1. There are a few details regarding Gastruloid generation that need clarification / justification since these steps deviate from conventional gastruloid culture protocols:

- a. ULA plates were used only for cells pre-cultured with higher levels of serum. Why? And why were different serum conditions used in first place? How does this affect the outcome variables? How does the use of ULA vs non-ULA plates affect outcome variables?*

The reason for testing the use of ULA vs non-ULA plates was to discard any potential

issue/difference during the process of aggregation (ULA plates force the cells to come together, facilitating aggregation). However, the reviewer is right in pointing out that we changed two variables at the same time (serum levels and ULA plates). For this reason, we performed new experiments isolating one of these variables (ULA plates). The results are in line with what was expected and can be observed in Fig. S1. Regarding the rationale for these experiments, we aimed to test whether variations in the pre-culture conditions, spanning those generally used by the labs working with gastruloids, could affect the differences observed between the use of homemade and commercial N2B27 media formulations - we made this point clear in the new version of our manuscript.

b. The outer wells of the 96-well plates were not used. What is the reason for that? How would using the outer wells affect the conclusions?

The outer wells were not used to ensure reproducibility because we noticed that sometimes there was higher evaporation in those wells, and gastruloid morphology was abnormal. This practice has also been used by Merle et al., 2024 (PMID: 38491138), and explored in higher detail in Mansouri et al., 2021 (PMID: 33855228).

c. The reported number of cells seeded per well was 200-300 cells. There have been multiple reports showing how gastruloid size affects their developmental dynamics (Bennabi et al., BioRxiv 2024; Fiuza et al., BioRxiv 2024; van den Brink et al., Development 2014; Anlas et al., Development 2024).

We agree.

d. Why are the gastruloids washed twice at 72h AA?

To avoid confusion, we have removed all the data obtained with an extra wash at 72h (Fig. S1A, E14Tg2A gastruloids - 2iL preculture). New data is now provided following the standard protocol.

2. Line 61: for initial number of cells, the authors should also cite Anlas et al., Development 2024 (PMID: 39552366).

We have included references for Anlas et al., 2024 and also Hamazaki et al., 2024 (PMID: 38405970) in the appropriate places of the new version of our manuscript.

3. Line 40-41: "cell fate decisions... coordinate cell movements and tissue morphogenesis" - this somewhat overstates the importance of cell fate decisions (relative to e.g. physical properties of the system).

We have modified this sentence as follows: "The development of an embryo is a highly organised process, with several factors such as signalling factors, gene and cell regulatory networks (GRNs and CRNs, respectively) and the physical constraints of the system regulating the allocation of distinct cell types and coordinating cell movements and tissue morphogenesis."

4. Line 58: what do the authors mean by "maintain their reproducibility"?

In this context, the phrase "maintain their reproducibility" means that over different experimental replicates, under the same conditions, the same frequency of observed phenotypes is constant within a small degree of error. Essentially, if we expect 90% of gastruloids to elongate, we expect to see something around 90% each and every time. We realise this statement needs clarification, and we have amended the text to the following: "It is therefore important to understand and properly characterise the factors that guide their differentiation, ensure reproducibility between experimental replicates, and acknowledge the variables that lead to suboptimal or alternative/variable phenotypes."

5. Line 62-63: what do the authors mean by "precision in the execution of the protocol"? Isn't precise execution of the protocol simply a manner of good scientific practice?

We removed this in the new version.

6. Line 111-112: "NDiff227 gastruloids display a more consistent shape from pole to pole". What do the authors mean by "more consistent shape"? What are the metrics measured to quantify this?

This essentially means that since there is a less pronounced anterior compartment, the anchoring circle of the elongation index is smaller in diameter, and so more of the total length of the gastruloid is considered part of the elongation, even if total length is the same in both conditions. The metrics to determine this are therefore those used in the Elongation index: circle diameter, and length. We have modified and clarified the text to read as follows: “*On the other hand, NDiff227 gastruloids display a more uniform shape from pole to pole (unlike HM- N2B27, which showed a larger anterior domain), which results in a larger elongation index*”.

7. *It is unclear to me what the exact relevance of using MEMs is in this context. The authors should better clarify this. To understand the analytical framework & assess its relevance and importance, it would be important to provide access to the Lunn et al. manuscript mentioned as "in preparation". I strongly recommend that the authors cite this as a preprint in the revised version of this manuscript.*

We have better described the relevance of using MEMs in the manuscript (see modified text). Regarding the Lunn et al. manuscript, this is indeed in preparation for the bioRxiv (and simultaneous submission elsewhere when complete). If the bioRxiv version is ready during typesetting of our manuscript, we will reference it. However, in lieu of this manuscript being available, we have chosen to cite a rather comprehensive paper which describes why MEMs are a better way to assess statistical significance when multiple variables could be interconnected - please see Yu et al. 2022 (DOI: 10.1016/j.neuron.2021.10.030). Nevertheless, we've added more information about the MEMs in the Materials and Methods section.

8. *Line 132-134: I don't understand the implied link between how the medium choice affects mentioned cell behaviours and cell competition. Do the authors suggest that there is less apoptosis in homemade N2B27?*

Yes, that was a strong possibility given the observed differences in cell number in the two types of gastruloids. We have investigated this issue in more detail and, as suggested by the reviewer, stained gastruloids for cleaved caspase 3 and phospho-histone H3 (pH3). The results (see Fig. 1F) indeed suggest there is likely more apoptosis in NDiff227 at 72h. Furthermore, we found no changes between the HM-N2B27 and NDiff227 conditions regarding pH3, suggesting that proliferation is similar in both media across the tested time-points.

9. *Line 152: this should be $p < 0.05$ as stated in Methods? Also, the Methods mention a logFC cut-off of 1, whereas the main text mentions a logFC cut-off of 1.5*

Yes, that was a typo - thank you for pointing it out. As for the log fold change, we have used a cut-off of 1.5 for visualization purposes in the volcano plot. A log fold change of 1 was chosen for the GO term analysis to enable a broader research on the variation between the gastruloids developed with the two media conditions. We have made this point clearer in the new version of our manuscript.

10. *Line 154: what is meant by "significant distance". I don't see a statistical analysis? "Significant" as in "considerable" - we made this alteration in the new version.*

11. *Line 195-197: could the authors distinguish between PGC-like cells & ectopic pluripotent cells?*

Please see our reply to point 7 raised by Reviewer #2.

12. *Line 204: since Hedgehog signalling is enriched in HM-N2B27 gastruloids, do the authors observe more endoderm and/or axial mesoderm?*

No major differences were noticed for these fates - we mention this in the main text (lines 230-233).

13. *Line 229:230: if the authors really want to make the point that the rate of cell proliferation, survival and/or differentiation is different, they should conduct experiments that show this (e.g. pH3, cleaved caspase stains).*

Following on from point 8 above, in addition to the cleaved caspase 3 stain, we also looked at pH3. Although there seems to be more cell death in NDiff227 at the earlier stages, pH3 does not significantly change between homemade and commercial media conditions. We

have discussed the meaning of this in the revised text.

14. Figure S3: distinct morphology needs to be quantified.

We have performed this quantification, and it is now in Fig. 2B.

15. For some metrics, e.g. elongation index, it seems that while the median is similar, the variance is different between conditions. The authors should comment on this.

We thank the reviewer for this comment, highlighting regions of the text where the MEM analysis could be clearer. We have modified the text (see manuscript) in order to clarify this, putting more emphasis on why the MEMs are utilised, and why they might be counterintuitive. A portion of the new text reads as follows “...For example, if gastruloid length were to increase, but the proportionality of the gastruloid remained consistent, area would also increase. Also, the EI is calculated as a ratio between the diameter of the largest inscribed circle within the gastruloid (which would relate to area) and the gastruloid total length (Fig. 1D) (Guet et al., 2021). To consider these variable interactions, which could obfuscate results and lead to misinterpreted conclusions (Yu et al., 2022) (and Lunn et al., in preparation), mixed effects models (MEMs) (Yu et al., 2022) were implemented to compare the morphologies of single axis gastruloids (10% ESL E14Tg2A) from both media...”

...and...

“Prior to the application of MEMs, observations of the raw data suggested that HM-N2B27 gastruloids have greater length and area than their NDiff227 counterparts, with no clear distinction in elongation index. However, after application of MEMs, HM-N2B27 gastruloids have a greater overall length ($p = 1.24 \times 10^{-4}$) whilst NDiff227 gastruloids have a greater elongation index ($p = 7.95 \times 10^{-4}$) and the observed difference in area is not significant (Fig. 1Cii, Fig. S2, & Table S1) (Guet et al., 2021)”

Second decision letter

MS ID#: dev.204774R1

MS Title: N2B27 media formulations influence gastruloid development

Authors: Tina Balayo; Sharna Lunn; Pau Pascual-Mas; Ulla-Maj Fiuza; Amruta Vasudevan; Joshua D. Frenster; Joel B. Josende García; Hannah Y. Galloon; Raquel Flores Peirats; Alfonso Martinez Arias; André Dias; David A. Turner

Article Type: Research Report

Dear Dr Turner,

I am happy to tell you that your manuscript has been accepted for publication in Development, pending our standard publication integrity checks.

Reviewer 1

SUGGESTIONS TO AUTHORS

Minor points:

- Cdx1 HCR panel is mislabelled in Figure 3. It should be Cdx2.

Reviewer 3

The authors should be commended for the thorough revision of the manuscript, including additional experiments, analyses, and textual clarifications. These further support and strengthen the main conclusions of the paper. I therefore recommend acceptance of this very important and timely study. I have just a few comments that the authors might want to take into consideration that would, in my view, further improve clarity & facilitate community implementation of the important findings of the study.

1. I understand why the authors have removed the comment regarding the study being conducted in two laboratories, but it comes at the cost of potential useful information for the community. Maybe the authors could comment in the Conclusion/Discussion section about critical parameters to ensure and examine gastruloid reproducibility across laboratories?

2. It would be very helpful for the community to add a Box with recommendations and guidelines for reproducible generation of gastruloids, and how this should be examined. Many of these are mentioned throughout the manuscript (e.g., fresh preparation of medium, shelf life, storage conditions of L-glutamine, exclusion of outer wells, etc), but in a scattered way, which makes it hard to distill critical information. Adding a Box with clear instructions for medium preparation, gastruloid generation, and subsequent QC would be a fantastic addition for the community.

3. In the rebuttal, the authors mention that, while all HM-N2B27 were fine, the commercial NDiff had good and bad batches. It would be good to mention this clearly in the main text (including % of batches good/bad), and clearly describe how this was assayed/decided (again, the latter could be part of the suggested Box).

4. I do not agree that the presence of a scale bar substitutes for through quantification of the two types of gastruloids shown in S1. For example, it does not substitute for quantification of elongation index. Adding similar quantifications as in 1B,C would further strengthen the paper.

5. I would highly recommend to add a citable version of Lunn et al., in preparation (MEM) prior to publication of the manuscript.